# North African humid periods over the past 800,000 years

Edward Armstrong [1] ✉, Miikka Tallavaara [1], Peter O. Hopcroft [2] & Paul J. Valdes [3,4]

The Sahara region has experienced periodic wet periods over the Quaternary and beyond. These North African Humid Periods (NAHPs) are astronomically paced by precession which controls the intensity of the African monsoon system. However, most climate models cannot reconcile the magnitude of these events and so the driving mechanisms remain poorly constrained. Here, we utilise a recently developed version of the HadCM3B coupled climate model that simulates 20 NAHPs over the past 800 kyr which have good agreement with NAHPs identified in proxy data. Our results show that precession determines NAHP pacing, but we identify that their amplitude is strongly linked to eccentricity via its control over ice sheet extent. During glacial periods, enhanced ice-albedo driven cooling suppresses NAHP amplitude at precession minima, when humid conditions would otherwise be expected. This highlights the importance of both precession and eccentricity, and the role of high latitude processes in determining the timing and amplitude of the NAHPs. This may have implications for the out of Africa dispersal of plants and animals throughout the Quaternary.

There is widespread palaeoclimatological evidence indicating that the Sahara has experienced wetter periods in the past, with proliferation of vegetation, rivers and lakes into what is now desert[1–16]. The alternation of arid and humid phases and consequent dramatic environmental changes in this vast region forms a key biogeographic factor influencing species' distributions and evolution[17–19]. It has also been suggested that changes in the precipitation and vegetation cover of the Sahara have consequences beyond Africa, influencing the climate system from the tropics to the Arctic[11]. Currently, 230 of these North African humid periods (NAHPs) have been identified in proxy records over the past 8 myr[8]. They were ultimately driven by the precession cycle, which governs the seasonal insolation contrast and is modulated by eccentricity, with periods of increased boreal summer insolation (i.e., precession minima) intensifying the African monsoon system[6,20–24]. Internal biogeophysical feedbacks then amplified the external orbital forcing and further enhanced precipitation[15,25–28].

Although precession is evident in numerous monsoon proxies[12,13,23], some records show gaps in NAHPs at intervals of precession minima termed skipped beats[12]. These coincide with glacial periods, suggesting that the monsoon response could be modulated by eccentricity and/or obliquity. These glacial skipped beats have previously been linked to the modulation of precession by eccentricity[7,8]. However the glacial boundary conditions, namely $CO_2$ and the extent and shape of the ice sheets which are additionally forced by eccentricity, may also indirectly influence the African monsoon response to insolation change[29,30]. Problematically, most climate models fail to reproduce the precipitation change expected during the NAHPs, including the most studied NAHP in the Holocene[31–33]. Furthermore, studies investigating the impact of glacial boundary conditions on the NAHPs are not conclusive. Some indicate that the NAHPs are not sensitive to the boundary conditions and are driven predominantly by orbital forcing[22,34]. Others conclude that NAHP suppression during the skipped beats was driven primarily by reduced

¹Department of Geosciences and Geography, University of Helsinki, Helsinki, Finland. ²School of Geography, Earth and Environmental Sciences, University of Birmingham, Birmingham, UK. ³School of Geographical Sciences, University of Bristol, Bristol, UK. ⁴Cabot Institute, University of Bristol, Bristol, UK. ✉e-mail: edward.armstrong@helsinki.fi

glacial $CO_2$ concentrations[6], or conversely was due to extensive ice sheets during glacials[9,15]. Another study concludes that both forcings were responsible for the skipped beats[21]. Currently therefore the mechanisms that drive the NAHPs remain poorly constrained.

Here, we address these outstanding research questions using a recently developed version of a coupled general circulation model (HadCM3BB-v1.0, see methods) that enables us to simulate the dynamics of NAHP events over the past 800 kyr. This allows us to clarify the forcings and mechanisms that drive the NAHPs and the skipped beats. Our results show that the modelled NAHPs are paced by precession due to their influence on the West African Monsoon (WAM) and regional jet streams. Glacial $CO_2$ concentrations do not significantly impact NAHP timing or amplitude. In contrast, we show that extensive ice sheets during glacials suppress NAHPs, this drives the skipped beats and elucidates a high latitude control on the NAHPs. Eccentricity therefore indirectly forces the NAHPs via its control over northern hemisphere glaciation. While the model simulates humid periods that have good agreement with regional proxy data across much of North Africa, precipitation levels remain below proxy-based expectations in the Northeast Sahara and the Arabian Peninsula, which suggests that our understanding of North African precipitation variability is possibly still incomplete.

## Results and discussion
### Simulating the NAHPs
Climate models largely fail to reproduce the intensification of NAHP precipitation implied by observations[31–33,35–38]. This disparity between models and observations has persisted for decades across numerous climate models, and has been linked to misrepresentation of vegetation, ocean and dust feedbacks[27,28,39,40]. How these could be reconfigured in models remains unclear[41,42]. Instead, recent work has modified convection and land-surface parameterisations[43,44]. This palaeo-conditioning was shown to markedly improve the simulation of the Holocene NAHP[43,44], however, it has not been tested beyond this time period.

Here we utilise the HadCM3B coupled climate model[45] and incorporate the updated atmospheric parameters and vegetation parameterisations identified by Hopcroft, et al.[44] in order to palaeo-condition the model and constrain NAHP timing and mechanisms over the past 800 kyr (methods). The model is herein termed HadCM3BB-v1.0. We performed 219 snapshot simulations with results splined to a continuous timeseries on a 100-yr timestep (methods). Although there are limitations to the snapshot approach (section 2, supplementary information), it has been shown to accurately simulate millennial palaeoclimate variability[46]. Moreover, it has advantages over accelerated approaches because it does not distort the underlying physics. Model validation (section 1, supplementary information) confirms that our methodology and HadCM3BB-v1.0 reproduces present-day and long-term patterns in global climate. Furthermore, HadCM3BB-v1.0 more accurately simulates mid-Holocene precipitation in the Sahara and other regions, not only relative to HadCM3B, but also against the majority of CMIP6 models[32] (Supplementary Fig. 1).

### Timing, amplitude and spatial pattern of the NAHPs
The timeseries for Saharan precipitation (15N:30°N, 15°W:35°E; Fig. 1i) indicates 20 NAHPs (Fig. 1h) over the past 800 kyr, defined as where precipitation exceeds one standard deviation above the mean (317 mm/yr). In all cases, the timing of the 20 NAHPs corresponds to precession minima (Fig. 1a), with 17 possible precession minima periods where the model simulates a skipped beat, although precipitation still increases during these periods compared to precession maxima (Fig. 1i, Supplementary Fig. 4).

The simulated NAHPs are compared with a range of proxy timeseries records (Fig. 1c–g) from across North Africa, selected due to their proximity to the study area, their long timescale, and high

temporal resolution. From the Western Sahara (Fig. 1d), the Zr/Rb ratio of terrestrial dust deposition derived from a North Atlantic marine core is indicative of aridity levels across the region[1]. From North East Africa, a humidity index (Fig. 1c) is derived from geochemical composition of a marine core from the Eastern Mediterranean, representative of monsoon runoff[2]. Similarly, sapropel intervals derived from three marine cores across the Mediterranean[47] (Fig. 1g), and a combined geochemical and principal component analysis of a different Mediterranean marine core[7] (Fig. 1f), are indicative of periods of enhanced moisture availability across North Africa. From the Gulf of Aden, the δD of leaf waxes derived from a marine core is a relatively direct proxy of precipitation amount in Northeast Africa[13] (Fig. 1e). The timing of possible analogous NAHPs are labelled in Fig. 1h and each of the proxy reconstructions. The modelled NAHP timing correlates well with those inferred from the observations. Potential observed NAHPs that are not simulated by the model are identified with a question mark in Fig. 1h, and the possible reasons for any model data discrepancy are discussed below.

Figure 2 shows the composite climatologies during periods of precession maxima (i.e., reduced boreal summer insolation), the 20 NAHPs identified in Fig. 1h, i, and the difference between the two periods. The average NAHP Saharan precipitation is 410 mm/yr (Fig. 1i). This varies spatially (middle panel; Fig. 2a, Supplementary Fig. 2), with west Saharan (15:30°N, 15°W:15°E) average NAHP precipitation of 552 mm/yr and maximum amplitude of 763 mm/yr (NAHP 6; Supplementary Fig. 2). Note that the model has a positive Saharan precipitation bias of 21 mm/yr (section 2 supplementary information) which may amplify these averages. In contrast to the western and central Sahara there is only a marginal increase in precipitation in Northeast Africa and the Arabian Peninsula (Fig. 2a), although the precipitation increase occurs contemporaneously in the Western and Eastern Sahara (Supplementary Fig. 2). The annual increase is dominated by anomalies in the summer months, with a small increase in winter precipitation in the Iberian Peninsula and Western Mediterranean, albeit not to the extent identified in other modelling studies (Supplementary Fig. 3)[6,21]. Proxy evidence for the Holocene NAHP indicates an increase ranging from ~400 mm/yr (15-30°N, 10°W:30°E)[48] to 640 mm/yr in the Western Sahara (15–33°N, 15°W:0°E)[14]. This compares to a modelled 494 mm/yr and 584 mm/yr for the same regions for 6–11 kyr BP.

During the NAHPs, HadCM3BB-v1.0 simulates the proliferation of forest to 18°N, and C4 grasses and shrubs across the central and western Sahara (Fig. 3a). When utilising a specific North African vegetation classification[8] (Fig. 3b), vegetation thresholds for sparse savannah grassland are estimated to be as little as 100 mm/yr, which is also the current lower boundary for Sahelian vegetation[49]. With this classification (Fig. 3b) the modelled NAHP precipitation sustains woodland to ~16°N, wooded grassland and grassland across the central and western Sahara, and southward migration of Mediterranean vegetation. Note that during the 17 skipped beats, precipitation increase is still significant enough to support the proliferation of grassland across the Western Sahara (Supplementary Fig. 4). This may influence proxy records and may be why some West Saharan dust and leaf-wax records indicate enhanced vegetation and precipitation at each precession minima[1,10,12].

The modelled NAHP amplitude in the North-eastern Sahara (Egypt and N.E. Libya), and the Arabian Peninsula is likely too small (Fig. 2a), with widespread evidence indicating regional river networks and lakes during the NAHPs[17,50–55]. In Northeast Africa, precipitation in the Tibesti mountains (360 mm/yr) and the upper Nile sources (Ethiopia, South Sudan, and southern Sudan) may have been enough to support river networks and lakes downstream. These, in addition to other remote sources such as groundwater, may have supported vegetation growth across the region, rather than enhanced local precipitation sources[19,50,56–58]. However, the Arabian Peninsula poses a more marked

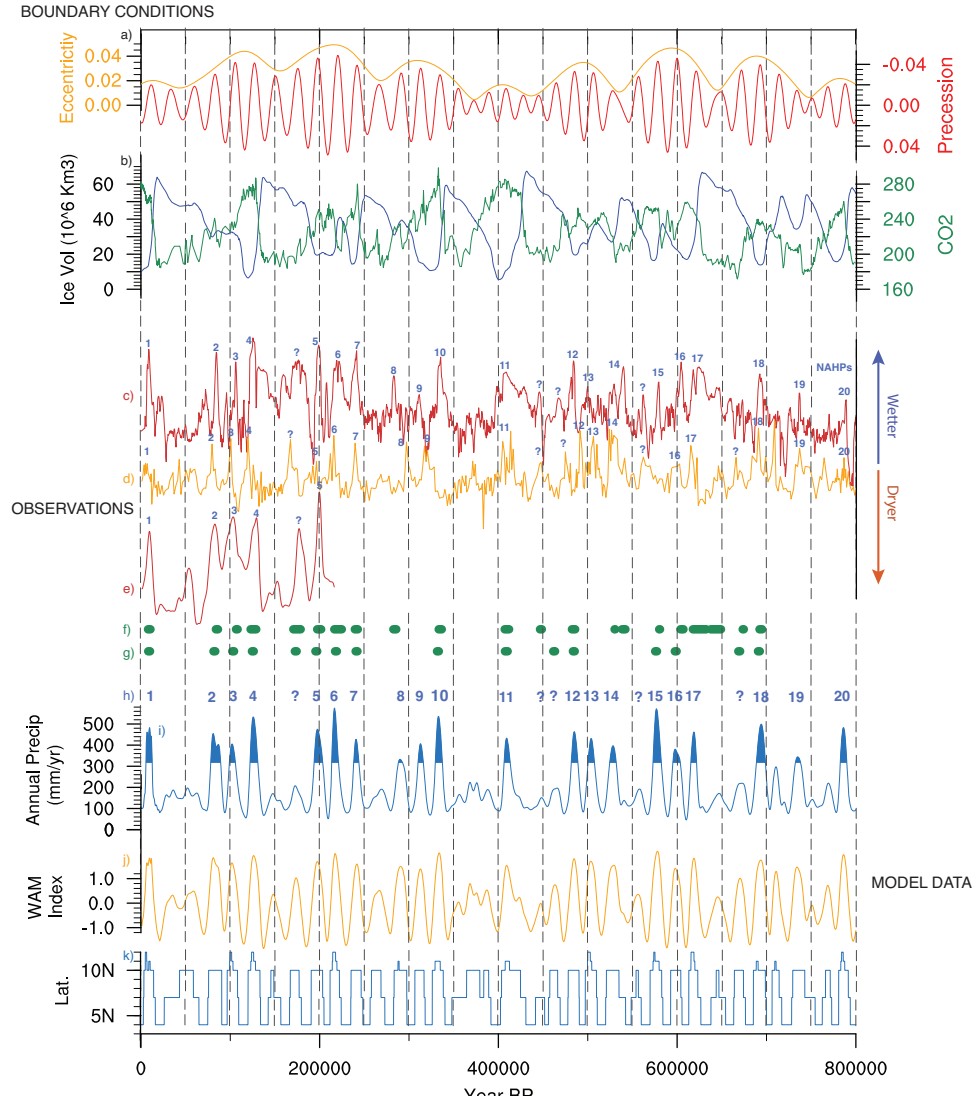

**Fig. 1 | Timeseries of the model boundary conditions, proxy observations of Saharan moisture availability and Mediterranean sapropels, and HadCM3BB-v1.0 model output for 0-800 kyr BP. a** Eccentricity and precession orbital parameters, **b** carbon dioxide ($CO_2$) concentration (ppm) and approximate ice volume ($10^6 km^3$) as calculated by the ice sheet model of de Boer, et al.[98], **c** North African humidity/aridity index derived from the Eastern Mediterranean[2] with potential humid periods labelled, **d** dust $\ln[Zr/Rb]$ ratio derived from a marine core off the Western Sahara[1], **e** $\delta D_{wax}$ data derived from the Gulf of Aden[13], **f** estimate of past North African Humid Period (NAHP) occurrence based on Eastern Mediterranean data[7], **g** stacked sapropel record which indicates humid phases derived from three cores (964, 966, 969) in the Eastern Mediterranean[47], **h** the numbered NAHPs identified from the model data (defined as where Saharan precipitation exceeds one standard deviation above the mean; 317 mm/yr) and possible NAHPs (denoted with a question mark) as indicated by observations, **i** modelled Saharan (15:30°N, 15°W:35°E) annual precipitation (mm/yr) with NAHPs marked, **j** a standardised West African Monsoon index[67], **k** latitude (°N) of maximum North African (−10:30°N, 15°W:35°E) summer (June, July, and August) precipitation.

conundrum. Modelled precipitation is limited to the south-west (580 mm/yr), which deteriorates to <100 mm/yr within the interior. We identify that this is an outstanding caveat in the model, and further research is required in order to identify the reasons for the model to underperform in this region, and how this should be rectified. It may be linked to an inaccurate response of the East African or Indian Monsoon systems to insolation change, or to the response of tropical plumes which affect Northern Arabia and may have been important during the NAHPs[59–62]. Additionally, Mediterranean storm tracks, which are an important winter moisture source along the Mediterranean coastline[53,63], may be poorly resolved in the model. Improved representation of these may enhance winter precipitation as shown in other modelling studies[6,21], albeit their impact was isolated to the coastlines. Further progress with palaeo-conditioning of the model, and utilising high-resolution or regional models which better resolve these processes[64–66], may reconcile these biases and is a focus of future work.

## Precession-driven mechanism of the NAHPs

Our results confirm that precession is the dominant pacemaker of the NAHPs. This is verified by wavelet analysis (Fig. 4d), which shows significant power at the precession period (~21 kyr) throughout the past 800 kyrs. The greater precipitation difference in the western Sahara indicates that the WAM is important in driving the modelled NAHPs in this region. In the Northeast Sahara and Arabian Peninsula where there is a much weaker modelled precipitation response, a different or additional driving mechanism may be prevalent. Periods of precession-driven increased summer boreal insolation (i.e., precession minima) shift the position of maximum precipitation and the Intertropical Convergence Zone (ITCZ; defined as the zone of trade wind

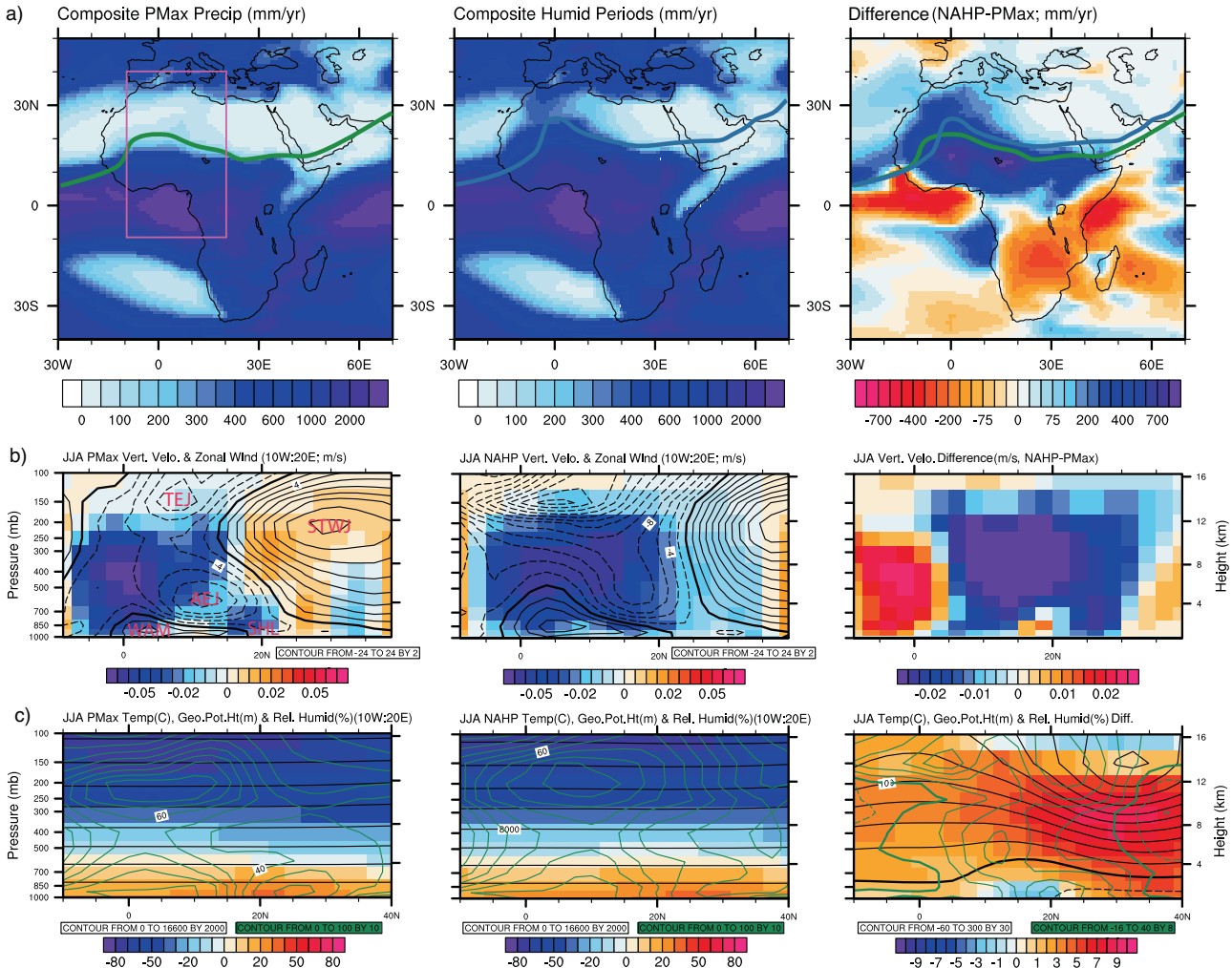

**Fig. 2 | Composite climatologies during periods of precession maxima (PMax), the 20 modelled North African Humid Periods (NAHPs), and the difference between the two climatologies. a** Annual average precipitation (mm/yr) and summer Intertropical Convergence Zone (ITCZ) position, defined as where the northern and southern trade winds converge. **b** West Saharan summer (June, July and August; JJA) latitude/height plots for the region identified in pink, showing lagrangian tendency of air pressure (filled colour contours; pa/s) where negative (positive) values represent upward (downward) motion, and zonal wind (black line contours; m/s) where solid/positive (dashed/negative) contours represent westward (eastward) air flow. The major components of the West African Monsoon are identified; SHL−Saharan heat low, AEJ and TEJ−African and Tropical Easterly Jets, STWJ−sub-tropical westerly jet. **c** West Saharan JJA latitude/height plots showing temperature (filled colour contours; °C), geopotential height (black line contours; m) and relative humidity (green line contours; %). Solid (dashed) contours represent positive (negative) differences. Only the differences that are considered 95% confident according to a student *t* test are shown (right panels).

convergence) northward when compared to periods of precession maxima (Figs. 1k and 2a), and this increases the strength of the WAM as indicated by the monsoon index (Fig. 1j; defined as the zonal speed of the African and Tropical Easterly Jets and low-level westerly winds averaged between 4:10°N and 20°W:20°E[67]).

The WAM is a complex system comprised of several key components (Fig. 2b)[68–70]. The summer monsoon, indicated by westerly (positive) zonal wind, extends to 18°N during precession maxima (Fig. 2b). Its northward penetration is dependent on the position of the anticyclonic Saharan Heat Low[71], atmospheric uplift here is not strong enough to cause precipitation. The rain belt is comprised of upward motion involving deep convection throughout the troposphere. The tropical air mass moves poleward forming the upper branch of the Hadley cell, during which it cools, descends and is deflected eastward by the Coriolis force generating the sub-tropical westerly jet (Fig. 2b). The Sahara is located below the descending branch of the Hadley Cell, where atmospheric subsidence inhibits convective rainfall. The African and Tropical Easterly Jets form in the middle and top of the troposphere, with the former sustained by dry-northerly and moist-

southerly convection[68,72,73]. The region between, and to the south in HadCM3BB-v1.0, of the African and Tropical Easterly Jets is characterised by strong uplift which favours precipitation[74] (Fig. 2b).

Observations indicate that the relationship between the African and Tropical Easterly Jets is tightly associated with Sahelian rainfall, with a more northerly weaker African Easterly Jet and/or stronger Tropical Easterly Jet associated with increased precipitation and vice versa[73,74]. Our results show that this same pattern is apparent during the NAHPs (Fig. 2b), in addition to an enhanced WAM that extends further north. The northward shift in the position of the ITCZ (Fig. 2a) is likely a response to a more positive north-south meridional temperature gradient during the NAHPs (Supplementary Fig. 5), a mechanism also implicated in enhanced Sahelian rainfall[75]. Furthermore, there is northward displacement, strengthening and warming of the sub-tropical westerly jet (Fig. 2b), with a positive mid-tropospheric geopotential height difference and increase in relative humidity (Fig. 2c). As a result, the descending air that characterises the sub-tropical westerly jet at precession maxima changes to ascending air during the NAHPs, indicating upper-level divergence and upward

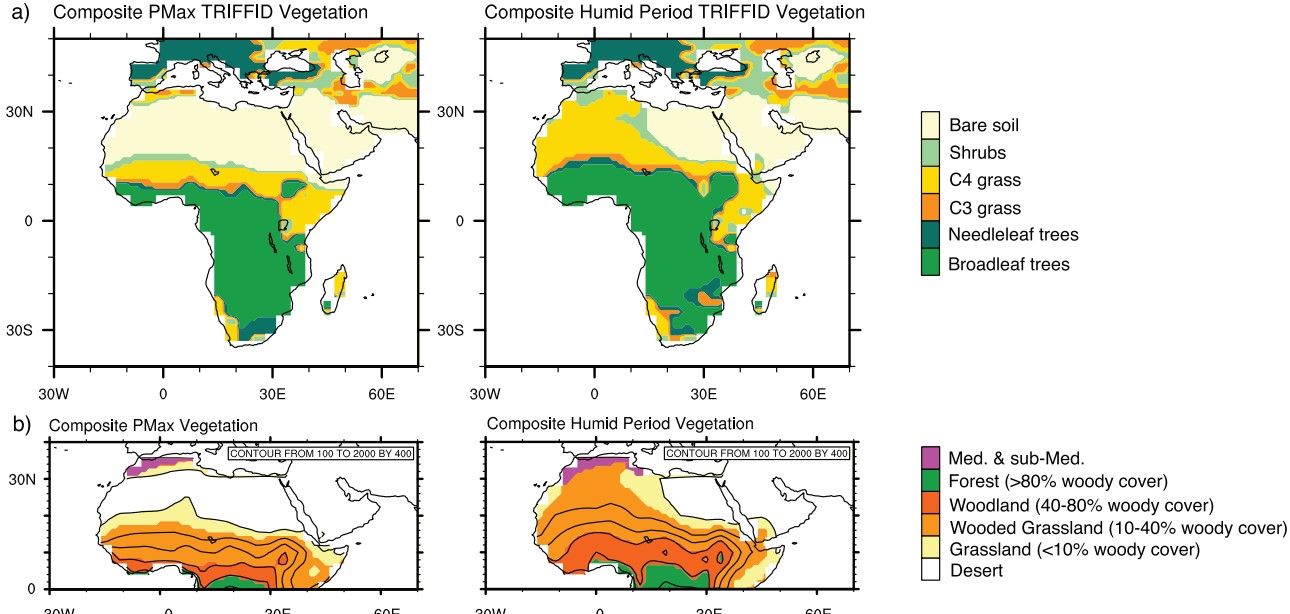

**Fig. 3 | Vegetation reconstructions during composite periods of precession maxima (PMax) and the 20 modelled North African Humid Periods. a** Output from the HadCM3BB-v1.0 dynamic global vegetation model (TRIFFID), showing only the most abundant plant function type in each grid square. **b** Vegetation reconstruction utilising the North African vegetation classification of Larrasoaña, et al.[8] which demonstrates regional vegetation types.

curvature of the jet forming a ridge over the Northern Sahara (Fig. 2b). It is the combined ITCZ, WAM, African/Tropical Easterly Jets, and the sub-tropical westerly jet response which results in increased precipitation in the Western and Central Sahara during the NAHPs. This initiates vegetation change and associated biogeophysical feedbacks including evapotranspiration which is dominant in the subtropics and tropics in HadCM3B[76]. This further enhances precipitation.

### NAHP forcing−orbits, greenhouse gases and the ice sheets

In addition to precession, a significant eccentricity signal and a weaker intermittent obliquity signal is evident in the wavelet analysis of modelled Saharan precipitation at dominant periods of 100 kyr and 41 kyr, respectively (Fig. 4d). Eccentricity may be present due to modulation of the precession cycle as has been previously proposed[7,8]. However, the signal may also highlight the influence of the glacial boundary conditions which fluctuate on these orbital timescales[77]. These forcings may influence the response of the monsoon system to precession and may consequently cause the skipped beats.

Past modelling studies that have investigated greenhouse gas and ice sheet forcing on monsoons and the NAHPs have proved inconclusive. A study[34] investigating the broad-scale response of the monsoon system to insolation change concluded that varying ice sheets and $CO_2$ had only a small impact on monsoon variability, which is dominated by precession forcing. A study using the CLIMBER-2 Earth system model of intermediate complexity (EMIC)[22] concluded that although glacial ice sheets and $CO_2$ concentrations contributed in suppressing NAHP precipitation during glacials, it was predominantly a response to weaker precession forcing (i.e., the modulation of precession by eccentricity). Kutzbach, et al.[6] used the CCSM3 model and concluded that glacial boundary conditions reduce summer NAHP precipitation, but the overall retreat of monsoon rain was relatively small. This was attributed to lower $CO_2$ concentrations, which cools tropical temperatures and reduces equatorial rains. Conversely, larger ice sheets increased winter precipitation due to a southward shift in the winter jet.

In contrast, studies using the LOVECLIM EMIC[9,15] emphasised the importance of ice sheet forcing. They concluded that ice sheet retreat at the last and penultimate glacials was required to initiate the Holocene[15] and Eemian[9] NAHPs. Timm, et al.[15] concluded that this retreat altered the background atmospheric state causing enhanced convective activity, this amplified the atmospheric response to orbital forcing. Menviel, et al.[9] concluded that glacial ice sheets caused a more southerly ITCZ and weaker WAM which inhibited AHP amplitude, although this was of secondary importance to the strength of the Atlantic meridional overturning circulation (AMOC). Both studies concluded that $CO_2$ concentration had no impact on AHP timing or amplitude. Another sensitivity study using LOVECLIM[21] also showed that NAHP supression during the last glacial was primarily due to ice sheet forcing, however they showed that $CO_2$ also influenced NAHP amplitude (see their Extended Figure 6), although the reasons for this were not explored. These studies therefore imply an ice sheet determined orbital signal in Saharan precipitation, however they all used a low resolution EMIC, covered a relatively short temporal timescale, and did not explore the direct mechanisms in detail[9,15,21].

In order to clarify the impact of the boundary conditions, we performed additional sensitivity experiments to evaluate the role of greenhouse gas and ice sheet forcing (Fig. 4a; methods). When incorporating only orbital variations (Orb_Only; Fig. 4a, b), Saharan precipitation is determined primarily by precession with a weaker obliquity signal (Fig. 4b). The absence of an eccentricity signal indicates two things. Firstly, that precession enhanced precipitation is not directly modulated by eccentricity, i.e., the magnitude of precession minima precipitation is not directly correlated with the extent of precession forcing, and correspondingly to its modulation by eccentricity. Secondly, that the eccentricity signal is therefore a response to the glacial boundary conditions. The addition of greenhouse gas forcing (Orb_GHG; Fig. 4a, c) has only a small impact on the amplitude and variability of Saharan precipitation. It is therefore the addition of the ice sheets (NAHP_All, Figs. 1i, 4a, d) that produces the strong eccentricity signal in Saharan precipitation and generates the skipped beats in the NAHPs. During glacial periods, extensive ice sheets suppress precession-induced precipitation change in the Sahara. Ice sheet volume is determined by eccentricity (Supplementary Fig. 6)[77] and therefore eccentricity indirectly forces the NAHPs via its influence on the ice sheets.

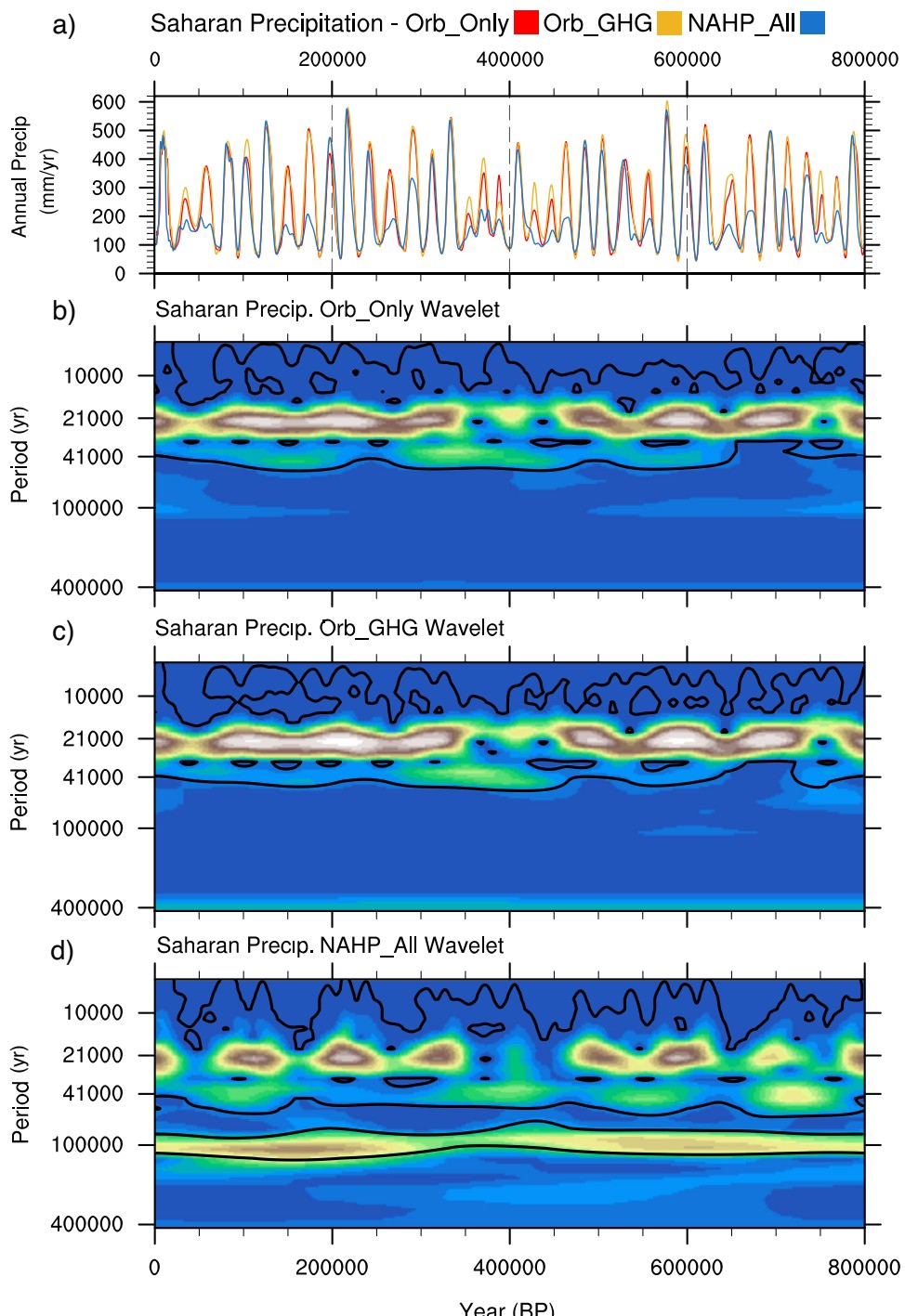

**Fig. 4 | Saharan precipitation timeseries and associated wavelet plots for the three forcing sensitivity experiments. a** Annual Saharan precipitation timeseries for the three forcing sensitivity experiments; Orb_Only, Orb_GHG, and NAHP_All. Orb_Only incorporates only orbital boundary conditions, Orb_GHG incorporates orbital and greenhouse gas boundary conditions, and NAHP_All incorporates orbital, greenhouse gas and ice sheet boundary conditions. **b–d** associated morlet wavelet power spectrum for the named experiment. The thick contour encloses regions with a 95% confidence level as determined by a red-noise process.

O'Mara, et al.[10] concluded that obliquity was a dominant control on NAHPs between 350 to 450 kyr BP. Our modelled results similarly indicate that obliquity has a significant influence on Saharan precipitation during this period (Fig. 4d). However, its longer-term influence over the past 800 kyr is secondary to precession and eccentricity. We also do not see a relationship between the strength of the AMOC and NAHP amplitude as identified in Menviel, et al.[9] (Supplementary Fig. 7). Furthermore, the important influence of ice sheets on NAHP amplitude, and uncertainties regarding their extent, growth and decay

phases[78], may explain the few inconsistencies between the modelled NAHPs and the observations as labelled in Fig. 1h.

Now that we have shown the key role of the ice sheets in controlling NAHP amplitude, it is important to identify how they suppress the NAHPs. Studies that have investigated the influence of ice sheets on monsoon evolution[79–81] identify both a thermodynamic and topographic impact. The former reflects albedo change, which alters absorbed insolation and consequently atmospheric and oceanic temperatures. For example sea surface temperatures have long been

implicated in the extent of Sahelian precipitation due to their control on the land–sea thermal contrast and WAM strength[82–85]. The topographic impact reflects ice sheet elevation, which influences synoptic scale wind patterns.

During glacial precession minima the ice sheets cool the atmosphere, decrease the meridional temperature gradient, weaken the WAM and cause the sub-tropical westerly jet to descend which inhibits precipitation (Supplementary Figs. 8 and 9), similar to periods of precession maxima (Fig. 2b). Using the atmosphere-only version of our model (HadAM3BB-v1.0; methods), we identify that this response is primarily due to the thermodynamic impact of the ice sheet. This is dominated by the atmospheric response to albedo change, with a smaller contribution from sea surface temperature changes (Supplementary Fig. 9; Section 3 supplementary information). During glacials, the increase in albedo and resultant colder atmosphere acts to decrease the global and Atlantic meridional temperature gradient, weaken the WAM, and results in a more southerly, weaker, colder and descending sub-tropical westerly jet, indicative of upper-level convergence and formation of a jet trough which inhibits convective precipitation (Supplementary Figs. 8 and 9). This is similar to the precession maxima mechanism. The glacial SSTs reduce precipitation by weakening the WAM, reducing the Atlantic meridional temperature gradient and shifting the position of the African/Tropical Easterly Jets (Supplementary Figs. 8 and 9; section 3 supplementary information). It is therefore a combination of the sea surface temperature and atmospheric response to albedo change that is the primary mechanism by which ice sheets suppress NAHPs during precession minima.

In summary, the ability of the HadCM3BB-v1.0 model to accurately simulate the timing and amplitude of the NAHPs in Western Africa confirms that palaeo-conditioning a model can improve palaeohydrological modelling in this region throughout the late Quaternary. The NAHPs were predominantly paced by precession and, in the west, by its influence on the WAM and sub-tropical westerly jet. In the Northeast Sahara and Arabian Peninsula, the model does not produce the NAHP response indicated by proxy data, suggesting an additional driving mechanism in this region. However, the model and proxies[1,2,13] (Fig. 1, Supplementary Fig. 2) produce a similar pattern in the timing and relative amplitude difference of the NAHPs in both the Western and Eastern Sahara. Furthermore, Northeast African proxies demonstrate a similar 100 kyr cyclicity to those in the Northwest, with humid phases coinciding with Northern Hemisphere interglacials[56,86–88]. These factors suggest that similar forcings are in effect across North Africa, even though the model may not produce the expected amount of precipitation in the Northeast. These uncertainties highlight the limitations of our understanding of the NAHPs, and demonstrate a continued need for future model development to clarify the mechanisms and resolve this disparity.

Our results show that during glacials, expansive ice sheets suppressed NAHP amplitude predominantly due to the impact of glacial albedo change, thus highlighting the role of high latitude processes in determining the timing and amplitude of the NAHPs. The ability to model NAHPs is important for understanding climate dynamics of the past and future, and enables the opportunity to study in much more detail the role of the Sahara as a factor controlling the dispersal of plants and animals, including humans, and the consequent movement of genetic and cultural information within and out of Africa.

## Methods
### HadCM3B model
The Hadley Centre Coupled Model 3 Bristol (HadCM3B) is a coupled climate model that consists of a 3d dynamical atmospheric component with a resolution of 2.5° × 3.75°, 19 vertical levels, and a 30 minute timestep[89], and an ocean model with resolution of 1.25° × 1.25°, 20 vertical levels and a 1-hour timestep[90]. Levels have a finer resolution towards the Earth surface. The model is a variant of HadCM3 that has

been developed at the University of Bristol, and is described in detailed in Valdes, et al.[45]. Despite its relative old age, the model has been shown to accurately simulate the climate system and remains competitive with more modern climate models. A key advantage of the model is its computationally efficiency, which permits long simulations and large ensemble studies. We also utilise the atmosphere-only version of the model; HadAM3B[45]. This incorporates the same atmospheric component as HadCM3B but with prescribed sea surface temperatures (SSTs).

The model incorporates the Met Office Surface Exchange Scheme (MOSES) version 2.1[91] which simulates water and energy fluxes and physiological processes such as photosynthesis, transpiration and respiration which is determined by stomatal conductance and consequently $CO_2$ concentration. The fractional coverage of nine surface types are incorporated by MOSES 2.1 and simulated by the dynamic global vegetation model (DGVM) TRIFFID. Of the nine surface types, five are plant functional types (PFTs); deciduous and needleleaf trees, C3 and C4 grasses, shrubs, with the residual assigned to bare soil. Vegetation evolves throughout a model simulation depending on temperature, moisture, $CO_2$ and competition with other PFTs. HadCM3B does not include an interactive carbon/methane cycle or ice model, so these boundary conditions have been imposed as discussed below.

### Palaeo-conditioning
The update to the atmospheric and vegetation parameters follows the work of Hopcroft, et al.[40,44]. The most important change was to modify the vertical profile of convective entrainment and detrainment, decreasing the entrainment rate near the surface and increasing the entrainment rate higher up in the atmosphere globally. This resulted in greater mixing between convective plumes and the environment in the upper troposphere relative to the lower troposphere increasing the precipitation response from African Easterly Wave events. Hopcroft, et al.[44] showed that this permitted the model to more accurately simulate mid-Holocene greening of the Sahara. At the same time, it did not adversely influence present-day climate simulations. The moisture stress function of vegetation was also altered to be linear in soil moisture potential rather than soil moisture, and then optimised to reproduce vegetation cover-climate relationships for the present day and the mid-Holocene[43].

The model was tuned using a Bayesian statistical methodology[92] targeted on seven observational targets. The performance of the model was quantified in a probabilistic sense that accounts for structural error in the model—that is the conditioning was specifically designed not to expect a perfect model—since this could easily lead to the right answer for the wrong reasons. Five of the tuning targets are for the present day: annual mean temperature, tropopause water vapour, central and north Africa annual mean precipitation and present-day tropical vegetation cover. The remaining two targets are derived from the mid-Holocene: pollen-inferred North African 6 kyr BP precipitation and vegetation cover. The paleo-conditioned model was then validated[43] using a leaf-wax precipitation record[14] and other records[93].

Together these updates allow a dynamic simulation of the Holocene greening of the Sahara that shows many similarities with the reconstructions of this time period[93], including a more gradual and later desertification in the East and South of the Sahara and a centennial-scale transition over North Western Sahara. The methodology differs to previous NAHP modelling studies[28,40,94] as they prescribed the greening of the Sahara, i.e., implicitly assumed that the Sahara is 100% vegetated, whereas here we are dynamically simulating the connection between forcings (e.g., $CO_2$, ice sheets, orbit) and the land-surface/atmosphere system.

African Humid Periods (and dry phases) that occurred before the Holocene were not used to condition (tune) the model in any way. This

study is therefore the first attempt to evaluate the palaeo-conditioned model for periods beyond the mid-Holocene against the spectrum of humid phases back to 800 kyr BP. We have incorporated the changes described above into HadCM3B, the updated model configuration is termed HadCM3BB-v1.0.

## Boundary conditions

The simulations have been forced with well-constrained orbital parameters[95] and greenhouse gas concentrations ($CO_2$, $N_2O$, and $CH_4$) taken from the Vostok Ice core[96,97].

The extent and elevation of the Antarctic, Greenland, North American and Fennoscandian ice sheets has been imposed utilising the reconstruction of de Boer et al.[98] This provides ice extent and thickness which are used within the model to calculate continental elevation (depending on ice thickness and isostatic rebound), bathymetry, ice sheet extent, and consequently the land–sea mask. There remain uncertainties regarding Ice sheet reconstruction beyond the LGM, in part due to poor preservation following the last deglaciation. Although the ice area might be approximated based on sea-level data, uncertainties remain associated with ice sheet structure during growth and decay phases. As stated in the text, this may cause uncertainty regarding the suppression of some the NAHP events and be the reason for inconsistencies between the model and reconstructions.

## Snapshot experiments

The boundary conditions have been incorporated into 219 snapshot experiments covering 0 kyr BP (corresponding to the year 1950 albeit with pre-industrial GHG concentrations) to 800 kyr BP. These are set at 1 kyr intervals for 0–24 kyr BP and 4 kyr intervals to for 24–800 kyr BP. Each simulation has been run for 500 years with analysis conducted on the final 50 years of each simulation unless stated otherwise. This spin-up permits the atmosphere and surface ocean to reach a state of near equilibrium. This approach allows the simulations to be run simultaneously and is therefore very efficient.

The snapshot climatologies were then splined to a 100-yr time-series for all variables used in the study, including the land–sea mask and ice fraction (subsequently rounded to 0 or 100% coverage in any grid cell). Splining has been done using the ftcurv function of the NCAR command language (NCL)[99] which utilises spline under tension. This is the same methodology to that outlined in Armstrong, et al.[46]. The timeseries data was then bilinearly interpolated to 1° resolution for surface variables. There has been no bias correction applied to the data in this study.

## Boundary condition sensitivity experiments

In order to identify the contributing role of three forcing factors on the NAHPs; orbital variations, GHGs and ice sheets, we performed two additional sets of snapshot experiments (2 × 219 simulations) using HadCM3BB-v1.0. The original set of snapshot simulations incorporate all three forcings (as discussed above) and is termed NAHP_All. In order to identify just the orbital forcing, the simulations were re-run with varying orbital parameters, but with constant pre-industrial GHGs ($CO_2$−280 ppm, $CH_4$−760 ppbv, and $N_2O$−270 ppbv) and ice sheet extent and elevation, this is termed Orb_Only. The third set of snapshot simulations incorporates both orbital and GHG forcing, with constant pre-industrial ice sheets, termed Orb_GHG.

The snapshot climatologies were similarly splined to a 100-yr timeseries and bilinearly interpolated to 1° resolution for surface variables.

## Topographic vs. thermodynamic ice sheet sensitivity experiments

In order to resolve the topographic vs thermodynamic contribution of the ice sheets to the suppression of the NAHPs during precession minima, we have used the atmosphere-only version of the model

(HadAM3B[45]) and incorporated the updates described above (HadAM3BB-v1.0). We have run the model controlling for SSTs, ice sheet elevation, and ice sheet albedo.

Due to computational limitations, we have not rerun the whole suite of snapshot simulations. Instead, we have performed a set of singular snapshot experiments corresponding to where the impact of the ice sheet is shown to have a significant suppressing effect on the amplitude of precession minima induced Saharan precipitation. We identify the eighth precession minima, corresponding to 176Kyr BP, as where ice sheet extent suppresses precipitation the greatest amount, from -510 mm/yr for Orb_Only/Orb_GHG to 205 mm/yr for NAHP_All (Fig. 4a). As the 176 kyr BP precipitation is equivalent for both Orb_Only and Orb_GHG, and because we only want to compare the impact of the ice sheet (rather than any possible impact of GHGs), we utilise input from the Orb_GHG experiment. We assume that our findings for this time period are consistent for all the suppressed NAHPs during glacials.

The first experiment is the control HadAM3BB-v1.0 simulation (termed Control). This incorporates the boundary conditions and SST input from the 176 kyr NAHP_All snapshot experiment. The climatology is therefore very similar to the 176 kyr BP HadCM3BB-v1.0 NAHP_All experiment (section 2, supplementary information). In order to isolate the thermodynamic impact of SSTs, we have run HadAM3BB-v1.0 with 176 kyr boundary conditions, but with SSTs prescribed from the 176 kyr Orb_GHG snapshot simulation, termed glacial_noSST. This simulation therefore has glacial forcings except for SSTs. To identify the thermodynamic impact of ice sheet albedo, we have run HadAM3BB-v1.0 with 176 kyr boundary conditions including SSTs, but with ice sheet albedo from the 176 kyr Orb_GHG experiment (this is equivalent to pre-industrial ice sheet albedo). This is termed glacial_noAlb. This simulation therefore has glacial forcings except for non-glacial (pre-industrial; PI) surface albedo. SSTs are fixed throughout the simulation so the albedo change will only impact the atmosphere. Note however that this is not a clean comparison because the height of the ice sheet will cause it to snow, so there will be a contribution from albedo in some locations. To identify the topographic impact of the ice sheet, we have run HadAM3BB-v1.0 with 176 kyr boundary conditions, but with ice sheet topography from the 176 kyr Orb_GHG experiment (this is equivalent to PI ice sheet topography). This is termed glacial_noTopo. This simulation has glacial forcings except for non-glacial (PI) topography. Each simulation was run for 200 years and analysis conducted on the final 30 years. An analysis of these simulations is provided in section 2 of the Supplementary Information.

## Data availability

A total of 661 model simulations were performed for this study. The raw model output and an overview of all the simulations is available at: https://www.paleo.bristol.ac.uk/ummodel/scripts/papers/Armstrong_et_al_2023.html. A document outlining how to access the NetCDF data files used in this study can be found at https://www.paleo.bristol.ac.uk/ummodel/scripts/papers/Using_BRIDGE_webpages.pdf. The relevant NetCDF file names for surface variables (precipitation, u and v wind components), full height atmosphere files (zonal wind, omega velocity, temperature, geopotential height, and relative humidity), and vegetation files are "[exptname]a.pdcl[month].nc", "[exptname]a.pccl[month].nc" and "[exptname]a.ptcl[month].nc" respectively, where [exptname] and [month] need to be defined by the user. The specific experiment names for each snapshot simulation are shown in the table in the provided link, and the month refers either to individual months ("jan", "feb", "mar" etc.), seasons ("djf", "mam", "jja", "son"), or the annual mean ("ann"). The CMIP6 mid-Holocene and PI control data is freely available at https://esgf-node.llnl.gov/search/cmip6/. Pollen reconstructions from Bartlein et al. (2011) and all proxy observation timeseries shown in Fig. 1 and the Supplementary Figs. are

downloadable as an electronic supplementary from the associated cited articles. ERA-5 reanalysis data is freely available from the ECMWF website at https://www.ecmwf.int/en/forecasts/dataset/ecmwf-reanalysis-v5.

## Code availability

All scripts used to analyse the data and produce the Figures have been written using the NCAR command language (NCL, Version 6.4.0) and are available from the GitHub repository https://doi.org/10.5281/zenodo.8261360.

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

## Acknowledgements

This work was carried out using the computational facilities of the Advanced Computing Research Centre, University of Bristol—http://www.bris.ac.uk/acrc (Bluecrystal). E.A. and M.T. acknowledge funding from the Kone Foundation (grant number 202006876). P.O.H. is supported by a University of Birmingham fellowship. Thank you to Frederik Schenk for the discussion on the topic area.

## Author contributions

E.A. conducted the analysis, and production of the figures and composed the original manuscript. M.T. contributed to the analysis and provided the observation datasets for comparison. P.O.H. and P.V. developed and published the palaeo-conditioning for the model. P.V. compiled and carried out the climate model simulations and the design of the experimental set-up. All authors commented on and reviewed the manuscript.

## Competing interests

The authors declare no competing interests.
