## [Peer Review File · Nature Communications]

North African Humid Periods over the past 800000 yearsReviewer #1 (Remarks to the Author):

Review of the paper "North African Humid Periods over the past 800000 years – Timing, Amplitude and Forcing" by Edward Armstrong and co-authors, submitted for publication in Nature Communications.

General comments

The paper presents an 800kyr climate dataset produced using a fully coupled climate model, used to study the past North African Humid Periods (NAHPs). The model simulates 20 NAHPs over the past 800kyr which have good agreement with the timing and amplitude of NAHPs identified in proxy data. The authors also detail the physical mechanisms controlling variability of the NAHPs. The paper is well written, in a clear and concise manner. The scientific background is well documented and the objective clearly stated. The methodology is appropriate and conclusions follow evidence. My overall assessment is entirely positive: results are convincing and represent an insightful and original progress in the field. I definitely recommend the paper for publication.

However I believe that, to improve the readability of the storyline, the authors should address some minor issues, mainly related with the presentation and the justification of the results. My specific comments are listed below.

Specific comments

In the introduction, it would be useful to spend a few words to make clear to the reader the link between precession and insolation.

L96: please make clear to the reader that the difference is between climatology of the precession maxima and NAHP climatology.

L131: please spend a few more words to help the reader make the connection with what is displayed in the plot and your interpretation.

L132: I'm not very convinced by the use you do of the word "anomaly", here and across the text. In my opinion, an "anomaly" is the difference with respect a climatology, while when you make the comparison between two composites, such in this case NAHP vs Pmax, using "difference" would be more appropriate. I'm not a native speaker, nor an expert in semantic, so I don't want to open a debate on this issue. However, if you find that this point makes sense, please check the use of "anomaly" across the text.

L134: please explain how ITCZ and WAMI are defined.

L149: this sentence depicts a too simplistic relationship between Sahelian rainfall and the jetstreams. Although it's true that a clear association exists between precipitation amounts and the Jetstream intensities and positions, this is not the only factor determining precipitation in the region (as you can find in all the references that you correctly cite here). I'd rather say that "Observations indicate that the relationship between the African and Tropical Easterly Jets is tightly associated with Sahelian rainfall..."

L157: not very clear to me what you mean with "jet ridge", please clarify.

L166: please indicate the (approximate) periodicities associated with eccentricity and obliquity.

L175: please define EMIC.

L192: please define MIS.

L195: please define AMOC. Also, please justify/explain how you conclude that AMOC doesn't influence NAHP.

L213-220: the explanation of the mechanism linking the ice sheets to the weakening of the WAM through the jet stream dynamics is too simplistic in my opinion. Although it's true that an increase in albedo leads to a colder global atmosphere, the change in the meridional thermal gradient also plays a major role in controlling the jet stream dynamics, and it would be a key ingredient in determining the variability of the WAM (see e.g. Biasutti M., 2019: Rainfall trends in the African Sahel: Characteristics, processes, and causes, *WIREs Clim. Change*, 10, e591, <https://doi.org/10.1002/wcc.59122> and references therein). Please rephrase this paragraph taking into account this aspect. An additional figure showing the thermal meridional gradients at the surface in your simulations would be helpful to better illustrate the mechanism.

Figure 1: some of the variables in the plot should be described. A dedicated section in the Methods could be helpful. In particular:

- a) Precession and eccentricity orbital parameters: what's the meaning of the values in the y-axis?
- c) Marine-isotope stages, please add a few words for non-experts in the field.
- d) North African humidity/aridity index derived from the Eastern Mediterranean: how is this index derived?
- e) Dust $\ln[\text{Zr}/\text{Rb}]$ ratio derived from a marine core off the Western Sahara: what does this ratio represent?
- f) $\delta\text{D}_{\text{wax}}$ data derived from the Gulf of Aden: what is the meaning of this index?
- k) A standardised West African Monsoon index: how is this index defined?

Figure 2: how is ITCZ defined? How can we identify statistically significant anomalies from non-significant ones? Same comment for Extended Figures 3, 4 and 6.

Extended Figure 6: check the titles for NAHP_All.

Supplement

L154: "which this".

L210-214: the impact of SST changes is not only associated with the modification of the regional land-sea thermal contrast, but also with the modification of the global meridional thermal gradient, and particularly in the North Atlantic (see e.g. Mohino, E., Janicot, S., Bader, J. (2011). Sahel rainfall and decadal to multi-decadal sea surface temperature variability. *Climate Dynamics*, 37(3–4), 419–440. <https://doi.org/10.1007/s00382-010-0867-2>). Please see also my comment to L213-220 in the main text.

Supplementary Figures 5 and 6 should be 4 and 5 instead, please check it out in the text.

Reviewer #2 (Remarks to the Author):

Key results

This study has three noteworthy features:

1. Use of a paleo-conditioned model: As the authors note, most climate models do not capture the full extent of the Holocene AHP. In this regard, using a climate model which has been 'tuned' to capture the Saharan greening is a good step and should be followed.
2. The long time scale studied: To the best of my knowledge, the longest previous modelling study was over the past 190 kyr (Duque-Villegas et al., 2022). The current study spans over the last 800 kyr and covers many more AHPs than the previous works. Further, some of the previous work was carried out using Earth Models of Intermediate Complexity, but the current work is based on a coupled General Circulation Model.
3. Identification of the importance of eccentricity: The long time span studied allows the authors to identify the eccentricity signal with more confidence than previous works (evident in their Figure 4d).

Originality and significance

Having identified some outstanding features of this study above, I'm not entirely convinced of its

novelty. The importance of precession in pacing the AHPs was already well-established. Further, the importance of eccentricity in modulating AHP magnitude is not entirely novel. Larrasoana et al. (2013) wrote, "Overall, we infer that WAM variability led to recurrent GSPs back to 8 Ma, with GSPs forming 400-kyr and 100-kyr clusters that attest to the impact of eccentricity modulation of precession (and hence of insolation) on monsoon variability ... The notion of eccentricity modulation of precession as the main driver of WAM variability provides strong support for a similar influence on the EAM .." Discussing Precession-caused High seasonality (PH) and Precession-caused Low seasonality (PL) Kutzbach et al. (2020) noted "The PH and PL extremes are largest when eccentricity is large and smallest when eccentricity is small during most of the glacial period." Thus, importance of precession, eccentricity or high latitude processes cannot be deemed the novelty of this work, however, it is true that this work provides robust evidence to support these points.

Validity

There are some points which the authors must address before the publication of this manuscript:

- Comparison with proxy data: The authors should organize and elaborate on model validation either in their response, or (preferably) in the manuscript. In Line 66, I note the reference: Hopcroft and Valdes (2021). In the section on "Paleoconditioning" in Supplementary Information, I note two references: Reference 41 - Hopcroft et al. (2021) and Reference 82 - Hopcroft and Valdes (2022). It would be helpful to have a specific description of where and how model validation was carried out, addressing – how many climatic parameters were assessed, how many archives and proxy records were used in the assessment and whether proxy-model agreement was quantified. On a first examination, it appears that only one leaf wax record and one biome reconstruction has been considered in these papers.
- Comparison with other studies of Holocene AHP: Notwithstanding the merits of the authors' approach to paleo-condition the model, it is not the only way to simulate a "reasonable" AHP, as shown by Pausata et al. (2016), Chandan et al. (2020) and Thompson et al. (2021). How does the paleo-conditioning approach used in the current study compare with the above mentioned studies in simulating the Holocene AHP?
- Significance of "Pmax": In Line 87, the authors note that the 20 AHPs identified correspond to precession minima. In Lines 93 and 135, Pmax or precession-maxima composite is referred to discuss NAHP Saharan precipitation. The authors should resolve this discrepancy, explain Pmax clearly in the main text and elaborate further on their observations regarding Figure 2.

References

- Line 46: Please elaborate on why Reference 6 (Kutzbach et al., 2020) was used to support the statement that some studies have concluded that ice sheet extent had little impact on AHPs.
- Line 172: Could the authors provide a brief explanation how the studies cited show that ice sheet is required for initiating an NAHP?
- The authors use two different citation styles in the main text. In most places, they use superscripts. However, in the last section, they use both superscripts and "ABC, et al."
- References 22, 71, 77 and 78 are incomplete. Reference 82 has a spelling mistake in the name of first author.

Clarity and context

There are a few points to address in the Abstract:

1. Line 16: The phrase "the position of the African monsoon system" is unclear and potentially misleading. The authors should replace it with something more commonly used in previous literature, like "extent" or "strength" instead of "position".
2. Line 17: Most IPCC-class climate models cannot generate enough precipitation to reconcile the magnitude of the Holocene AHP; there is no coordinated PMIP experiment for the previous AHPs.
3. Line 21: I think "confirm" is a strong word since it is well established (and not contested) that precession paces the AHPs. The reported results are, instead, in alignment with what is already known.
4. Line 23: "glacial periods".
5. Line 26: The last sentence is out of place. This is not because it is untrue, but because it is completely outside the purview of this study. In fact, the potential implications for the out of Africa

dispersal are mentioned only once in the main text, in the last line alone.

Suggested improvements

The manuscript needs careful proofreading and some improvements for greater readability. Some suggestions are listed below, but the list is likely not exhaustive:

- Line 38: Inconsistent tense (should use "amplified" and "enhanced")
- Line 47: I believe the authors want to convey that the mechanisms of the NAHPs are poorly constrained, not our ability to understand them.
- Line 76: Please indicate a section in the Supplementary Information as "Model validation".
- Line 143: Please elaborate on the connection between the subtropical westerly jet and the meridional Hadley Circulation.
- Lines 159 and 160: The authors should avoid using pronouns for subjects. In Line 159, biogeophysical feedbacks are not enhanced but activated / initiated.
- Line 183: Perhaps the authors want to convey that the magnitude of precession-minima precipitation is not directly correlated with eccentricity forcing.
- Line 194: There is no Figure 3f in this manuscript.
- Figure 2: Pmax is indicated in the left panel.
- Figure 4, Extended Figures 3, 4 and 5: Please rephrase the last sentence of the caption.
- All through the manuscript, please use 15 °W instead of -15 °E (for example).

Reviewer #3 (Remarks to the Author):

Armstrong et al. present a well-written and potentially critical study on North African humid periods (NAHP) during the past 800,000 years. The overall approach applied by the authors is exciting, but I remain unsure if the results can be considered to be robust. For the sake of transparency, I need to say that I am not a modeler, and thus, reviewing this paper from the perspective of a "proxy-person," not a modeler, which also means that some modeling details are not necessarily apparent to me.

The reason I mainly struggled with this paper is threefold:

Firstly, the model reproduces modern precipitation patterns well enough (Supplementary Figure 1), but it tends to produce more widespread precipitation patterns that are more intense than the compared to the average observation time series (1950-2020). So, it seems the model is always on the "too wet" side for regions it produces precipitation, thus potentially overrepresenting wet phases in some regions.

At the same time, the model does not represent specific precipitation patterns shown to have existed during past NAHP based on proxy records. That is particularly true for the entire North-Eastern Sahara region. In all fairness, this model and proxy data mismatch for this region is recognized by the authors (Line 116-127) but not further discussed in detail within the discussion. In my mind, however, this represents a huge cavity of the model. From the way it looks to me, the model does not only underrepresent the precipitation in North-Eastern Africa, as stated by the authors, but it simply does not produce precipitation for half of North Africa even though from proxy data, substantial lakes and active river system have been reconstructed for this region (e.g., Blanchet et al., 2021; <https://doi.org/10.1038/s41561-020-00671-3>).

Lastly, the authors claim that the model overall produces the NAHP well across the past 800 kyr, including the "skipped beats." However, I need help seeing this proposed match in the Fig. 1j record. Most of the skipped beats are expressed as "glacial sapropels" in proxy records of the Mediterranean and thus are reflected to some extent in the wet-dry index of the Mediterranean Sea (Fig. 1d). Yet, the model, in my view, persistently fails to produce enough precipitation for these time periods. Based on Fig. 1, I am not convinced the model works well for these time periods.

Put all the above together, I need help understanding how the model output can be used to assess North African humid periods if the model does not reproduce half of North Africa particularly well

while the other half might be overrepresented due to the sensitivity of the model. Additionally, some threshold conditions are seemingly not presented. Hence, I would appreciate it if the authors could revise the manuscript to address these issues, which seems crucial to get a feeling of the robustness of the model output. Currently, and I am sorry to say this, but the output does not convince me.

Generally, the authors could incorporate available proxy records much more. While the authors use proxy records to assess model robustness for surface air and sea surface temperatures (Supplementary Figures 2 and 3), the cross-validation with proxy records in terms of the precipitation output is not existing. The authors use the wet-dry index from the Mediterranean, the dust record from the West Sahara region, and the dD record from the Gulf of Aden to identify the NAHP. But why these records? Why not use Lake Bosumtwi for West Africa for some of the studied period (Miller et al., 2016; <https://doi.org/10.1002/jqs.2893>)? Why not use Chew Bahir, Lake Magadi, or Lake Tana proxy data to assess NAHP from the eastern African viewpoint (e.g., Foerster et al., 2022; <https://doi.org/10.1038/s41561-022-01032-y>)? The authors used the data for temperature. Why not precipitation? Why use the Gulf of Aden proxy record when it is so short? The selection of records seems random, and I would like the authors to explain why they selected those records and how they can be meaningful to the reader.

Regarding the vegetation output, I would also like to invite the authors to provide cross-evaluation based on proxy data. This, too, could be cross-checked by using the NEOTAMA database and CREST (Chevalier et al., 2022; <https://doi.org/10.5194/cp-18-821-2022>). While pollen data for the past 120 kyr is more readily available than for the past 800 kyr, the composite vegetation pattern produced from the 20 NAHP could be cross-referenced with what is available in terms of proxy data.

Lastly, the authors state that winter storm tracks, a vital moisture source during winter to North Africa (L116-127), need to be better resolved in the model. The authors still go on to discuss high-low latitude atmospheric teleconnections to explain the precipitation patterns the model provides. How can the mechanism be discussed if an important atmospheric aspect linking high and low latitudes and which is also sensitive to European Ice sheet conditions (e.g., southward extension; Luetscher et al., 2015; <https://doi.org/10.1038/ncomms7344>) is not represented in the model? I would appreciate it if the authors could discuss this aspect. How would that change the results, assuming that storm tracks have an effect? Why does it not matter that the storm tracks are not represented?

In summary, I am intrigued by the author's findings that CO₂ and AMOC changes did not contribute much to the precipitation patterns across Africa. This builds on recent proxy results suggesting something similar (Gosling et al., 2021; <https://doi.org/10.1126/science.abg4618>). The separation of the short eccentricity cyclicity from the precession cyclicity and the subsequent argument that the former stems from Ice sheet variability rather than the low latitude insolation forcing itself is a significant finding. However, I am unsure about the model output as a whole; hence the wavelet results are also difficult to judge for me at this point.

Reviewer #4 (Remarks to the Author):

Co-reviewed with reviewer 2

Overview of Manuscript Corrections

In response to the reviewer comments, we have made the following major changes to the manuscript:

- We have added a precipitation validation to section 1 of the supplementary information. This gives a comprehensive overview of 16 African precipitation proxies and compares against our model data. In most instances, the precipitation observations show good agreement with the model data which demonstrates the robustness of our model results.
- We have added a paragraph to Section 3 of the manuscript justifying our choice of proxies that we have used in Figure 1.
- We have acknowledged that the lack of increased precipitation in NE Africa is a caveat of the model and expanded the discussion as to why this might be the case.
- We have discussed the role of the meridional temperature gradient, including how changes to this contribute to the NAHP mechanism and how this is affected by the addition of the ice sheets.
- We have expanded the discussion on previous studies that have investigated the impact of the boundary conditions on the amplitude of the NAHPs. We have highlighted that the hypothesis that ice sheets may suppress the NAHPs has been previously proposed, but that past modelling studies have been inconclusive. Our study provides robust evidence in support of their impact on NAHP suppression.
- We have expanded the discussion on the palaeo-conditioning of the model in the Methods.
- We have made a number of changes to the figures, and added two additional figures to clarify the impact of the meridional temperature gradient (EF5/EF9) and the impact of AMOC forcing (EF7).
- We have made numerous corrections to the manuscript and figures as recommended by the reviewers, and as shown in the tracked changes. A detailed point by point response is shown below.

REVIEWER COMMENTS

Reviewer #1 (Remarks to the Author):

Review of the paper "North African Humid Periods over the past 800000 years – Timing, Amplitude and Forcing" by Edward Armstrong and co-authors, submitted for publication in Nature Communications.

General comments

The paper presents an 800kyr climate dataset produced using a fully coupled climate model, used to study the past North African Humid Periods (NAHPs). The model simulates 20 NAHPs over the past 800kyr which have good agreement with the timing and amplitude of NAHPs identified in proxy data. The authors also detail the physical mechanisms controlling variability of the NAHPs. The paper is well written, in a clear and concise manner. The scientific background is well documented and the objective clearly stated. The methodology is appropriate and conclusions follow evidence. My overall assessment is entirely positive: results are convincing and represent an insightful and original progress in the field. I definitely recommend the paper for publication.

Thank you for your helpful review comments.

However I believe that, to improve the readability of the storyline, the authors should address some minor issues, mainly related with the presentation and the justification of the results. My specific comments are listed below.

Specific comments

In the introduction, it would be useful to spend a few words to make clear to the reader the link between precession and insolation.

We have now added some text highlighting the link between precession, insolation and its modulation by eccentricity.

L96: please make clear to the reader that the difference is between climatology of the precession maxima and NAHP climatology.

We have clarified this in the text

L131: please spend a few more words to help the reader make the connection with what is displayed in the plot and your interpretation.

We have expanded the sentence in the text.

L132: I'm not very convinced by the use you do of the word "anomaly", here and across the text. In my opinion, an "anomaly" is the difference with respect a climatology, while when you make the comparison between two composites, such in this case NAHP vs Pmax, using "difference" would be more appropriate. I'm not a native speaker, nor an expert in semantic, so I don't want to open a debate on this issue. However, if you find that this point makes sense, please check the use of "anomaly" across the text.

We agree. As PMax and the NAHPS (PMin) are 'extreme' end member climatologies then 'difference' does seem to be more appropriate here. We have changed this word throughout the text and figure captions.

L134: please explain how ITCZ and WAMI are defined.

We have defined these in the text.

L149: this sentence depicts a too simplistic relationship between Sahelian rainfall and the jetstreams. Although it's true that a clear association exists between precipitation amounts and the Jetstream intensities and positions, this is not the only factor determining precipitation in the region (as you can find in all the references that you correctly cite here). I'd rather say that "Observations indicate that the relationship between the African and Tropical Easterly Jets is tightly associated with Sahelian rainfall..."

We have changed the wording to what you have recommended

L157: not very clear to me what you mean with "jet ridge", please clarify.

We have clarified this sentence in the text.

L166: please indicate the (approximate) periodicities associated with eccentricity and obliquity.

Ok we have defined the time periods in the text.

L175: please define EMIC.

We altered the text here and expanded out the acronym.

L192: please define MIS.

We have now removed the mention of MIS here as we do not use it elsewhere in the manuscript and instead explicitly stated the years that they identify obliquity in their wavelet plot. We have also removed this from Figure 1.

L195: please define AMOC. Also, please justify/explain how you conclude that AMOC doesn't influence NAHP.

We have defined the AMOC and added Extended Figure 7 which shows that there is no relationship between AMOC strength and timing of NAHPs

L213-220: the explanation of the mechanism linking the ice sheets to the weakening of the WAM through the jet stream dynamics is too simplistic in my opinion. Although it's true that an increase in albedo leads to a colder global atmosphere, the change in the meridional thermal gradient also plays a major role in controlling the jet stream dynamics, and it would be a key ingredient in determining the variability of the WAM (see e.g. Biasutti M., 2019: Rainfall trends in the African

Sahel: Characteristics, processes, and causes, WIREs Clim. Change, 10, e591, <https://doi.org/10.1002/wcc.59122> and references therein). Please rephrase this paragraph taking into account this aspect. An additional figure showing the thermal meridional gradients at the surface in your simulations would be helpful to better illustrate the mechanism.

This is a good point. We have expanded upon this in a number of locations throughout the manuscript and supplementary. We have added Extended Figure 5 which shows a timeseries of the global meridional temperature gradient with Saharan precipitation, showing the link between the NAHPs and increased meridional temperature gradient. We have discussed this Figure in the final paragraph of Section 4. We have also added to Extended Figure 9 showing both the global and Atlantic meridional temperature gradients for each of the sensitivity experiments. These are discussed in the final section of the manuscript and in more detail in Section 2 of the supplementary information.

Figure 1: some of the variables in the plot should be described. A dedicated section in the Methods could be helpful. In particular:

a) Precession and eccentricity orbital parameters: what's the meaning of the values in the y-axis?

These are dimensionless parameters, therefore the y-axis values do not have a unit.

c) Marine-isotope stages, please add a few words for non-experts in the field.

We have decided to remove this as we do not refer to it in the text.

d) North African humidity/aridity index derived from the Eastern Mediterranean: how this index is derived?

e) Dust $\ln[\text{Zr}/\text{Rb}]$ ratio derived from a marine core off the Western Sahara: what does this ratio represent?

f) dD_{wax} data derived from the Gulf of Aden: what is the meaning of this index?

We have added an additional paragraph to Section 3 which gives a more detailed overview of our choice of proxies in Figure 1, how they are derived and what they represent.

k) A standardised West African Monsoon index: how is this index defined?

We have now defined this in the text.

Figure 2: how is ITCZ defined? How can we identify statistically significant anomalies from non-significant ones? Same comment for Extended Figures 3, 4 and 6.

The ITCZ is defined as the position where the Northern and Southern trade winds converge – we have added this to the Figure caption. Only the differences that are considered 95% are shown. We have altered the text here and for the other appropriate Figures.

Extended Figure 6: check the titles for NAHP_All.

We have changed this.

Supplement

L154: "which this".

We have corrected this.

L210-214: the impact of SST changes is not only associated with the modification of the regional land-sea thermal contrast, but also with the modification of the global meridional thermal gradient, and particularly in the North Atlantic (see e.g. Mohino, E., Janicot, S., Bader, J. (2011). Sahel rainfall and decadal to multi-decadal sea surface temperature variability. *Climate Dynamics*, 37(3–4), 419–440. <https://doi.org/10.1007/s00382-010-0867-2>). Please see also my comment to L213-220 in the main text.

We agree. We have adapted Extended Figure 9 to show both the global and the Atlantic meridional temperature gradients for the different sensitivity experiments. We have expanded the text in Section 2 of the supplementary to discuss the impact of the meridional temperature gradient and added the references you recommend. We have also adapted the manuscript as discussed above.

Supplementary Figures 5 and 6 should be 4 and 5 instead, please check it out in the text.

This has been corrected.

Thank you for your review comments.

Reviewer #2 (Remarks to the Author):

Key results

This study has three noteworthy features:

1. Use of a paleo-conditioned model: As the authors note, most climate models do not capture the full extent of the Holocene AHP. In this regard, using a climate model which has been 'tuned' to capture the Saharan greening is a good step and should be followed.
2. The long time scale studied: To the best of my knowledge, the longest previous modelling study was over the past 190 kyr (Duque-Villegas et al., 2022). The current study spans over the last 800 kyr and covers many more AHPs than the previous works. Further, some of the previous work was carried out using Earth Models of Intermediate Complexity, but the current work is based on a coupled General Circulation Model.
3. Identification of the importance of eccentricity: The long time span studied allows the authors to identify the eccentricity signal with more confidence than previous works (evident in their Figure 4d).

Thank you for your review comments.

Originality and significance

Having identified some outstanding features of this study above, I'm not entirely convinced of its novelty. The importance of precession in pacing the AHPs was already well-established. Further, the importance of eccentricity in modulating AHP magnitude is not entirely novel. Larrasoña et al. (2013) wrote, "Overall, we infer that WAM variability led to recurrent GSPs back to 8 Ma, with GSPs forming 400-kyr and 100-kyr clusters that attest to the impact of eccentricity modulation of precession (and hence of insolation) on monsoon variability ... The notion of eccentricity modulation of precession as the main driver of WAM variability provides strong support for a similar influence on the EAM .." Discussing Precession-caused High seasonality (PH) and Precession-caused Low seasonality (PL) Kutzbach et al. (2020) noted "The PH and PL extremes are largest when eccentricity is large and smallest when eccentricity is small during most of the glacial period." Thus, importance of precession, eccentricity or high latitude processes cannot be deemed the novelty of this work, however, it is true that this work provides robust evidence to support these points.

We agree that the role of precession is well-established and we state this in the manuscript. We also agree that past studies have highlighted a role of eccentricity in the timing and/or amplitude of the NAHPs. We have cited these in the introduction, including Skonieczny, et al. (2019) who identified the role of eccentricity in determining the skipped beats. However, past studies such as the ones you quote above, have linked the impact of eccentricity on the amplitude of the NAHPs due to its *modulation* of precession. This is the case for both the Larrasoña et al. (2013) and Kutzbach et al. (2020) quotes that you provide above.

However in this study, we are showing that although eccentricity does modulate precession, this does not translate to an eccentricity signal in the NAHPs when the orbital signal is assessed alone (as shown in Figure 4b). Instead, the eccentricity signal in the NAHPs is determined by its control over the extent of the ice sheets. Therefore, eccentricity influences the amplitude of the NAHPs and their suppression via its indirect control over the extent of the ice sheets, which is the primary driver of the skipped beats during glacials.

The link between glacial ice sheet extent and the amplitude of the NAHPs has also been identified and discussed in past modelling studies using the LOVECLIM EMIC. However, other studies have come to different conclusions. We have expanded the discussion of these studies in Section 5, highlighting that this hypothesis has been previously proposed but remains uncertain. Although we are not the first to identify the impact of this forcing, this is the first study to use a more complex GCM over a much longer time period covering numerous glacial/interglacial cycles. Therefore as you state, it provides robust evidence in support of this hypothesis. We also provide a detailed mechanism for how the ice sheets suppress the NAHPs during glacials, which to our knowledge, has not been investigated in previous studies.

Validity

There are some points which the authors must address before the publication of this manuscript:

- Comparison with proxy data: The authors should organize and elaborate on model validation either in their response, or (preferably) in the manuscript. In Line 66, I note the reference: Hopcroft and Valdes (2021). In the section on "Paleoconditioning" in Supplementary Information, I

note two references: Reference 41 – Hopcroft et al. (2021) and Reference 82 – Hopcroft and Valdes (2022). It would be helpful to have a specific description of where and how model validation was carried out, addressing – how many climatic parameters were assessed, how many archives and proxy records were used in the assessment and whether proxy-model agreement was quantified. On a first examination, it appears that only one leaf wax record and one biome reconstruction has been considered in these papers.

We have added a precipitation validation in section 1 of the supplementary information which gives a comprehensive overview of African precipitation proxies and compares against our model data, in addition to the SAT and SST validation. In most instances, the precipitation observations show good agreement with the model data which clarifies the robustness of our model results.

With regards to the paleoconditioning, we have added an additional paragraph to the Methods section which discusses in greater detail how the paleoconditioned model in the studies of Hopcroft et al. (2021) & Hopcroft and Valdes (2022) was tuned and validated. We have kept this to one paragraph, as the methodology is fully documented and already published, along with associated code, in the provided references. We have added the additional reference to the first paragraph of section 2.

- Comparison with other studies of Holocene AHP: Notwithstanding the merits of the authors' approach to paleo-condition the model, it is not the only way to simulate a "reasonable" AHP, as shown by Pausata et al. (2016), Chandan et al. (2020) and Thompson et al. (2021). How does the paleo-conditioning approach used in the current study compare with the above mentioned studies in simulating the Holocene AHP?

All three of the studies cited *prescribe* the greening of the Sahara (Pausata et al. (2016), Chandan et al. (2020) and Thompson et al. (2021)). This means that they are not able to clarify how the climatic forcings translate to the climatic/environmental results because they implicitly assume that the Sahara is 100% vegetated. These studies therefore only show how the rainfall responds when subject to a prescribed land-atmosphere feedback. Our work differs because we are dynamically simulating the connection between forcings (e.g. CO₂, ice-sheets, orbit) and the land-surface/atmosphere system. We have stated this in the Methods.

- Significance of "Pmax": In Line 87, the authors note that the 20 AHPs identified correspond to precession minima. In Lines 93 and 135, Pmax or precession-maxima composite is referred to discuss NAHP Saharan precipitation. The authors should resolve this discrepancy, explain Pmax clearly in the main text and elaborate further on their observations regarding Figure 2.

We have used the PMax composite to show the average precipitation during opposing precession periods, i.e. during periods of decreased summer boreal insolation. We have clarified this in the text and explained in more detail what this Figure shows in the third paragraph of this section.

References

- Line 46: Please elaborate on why Reference 6 (Kutzbach et al., 2020) was used to support the statement that some studies have concluded that ice sheet extent had little impact on AHPs.

We have reworded this section and added two additional paragraphs in Section 5 which expand upon the conclusion of this, and the other boundary condition studies.

Kutzbach concluded that the glacial ice sheets increased winter rainfall, and glacial CO₂ levels decreased summer rainfall. However, the overall impact of the boundary conditions on monsoon rain during glacials was small.

- Line 172: Could the authors provide a brief explanation how the studies cited show that ice sheet is required for initiating an NAHP?

We have expanded the discussion of all these studies in paragraphs 2 and 3 of this section.

- The authors use two different citation styles in the main text. In most places, they use superscripts. However, in the last section, they use both superscripts and "ABC, et al."

We have confirmed with the editor that both citation styles are ok to use.

- References 22, 71, 77 and 78 are incomplete. Reference 82 has a spelling mistake in the name of first author.

Thanks for identifying these, we have corrected them.

Clarity and context

There are a few points to address in the Abstract:

1. Line 16: The phrase “the position of the African monsoon system” is unclear and potentially misleading. The authors should replace it with something more commonly used in previous literature, like “extent” or “strength” instead of “position”.

Ok we replaced ‘position’ with ‘intensity’.

2. Line 17: Most IPCC-class climate models cannot generate enough precipitation to reconcile the magnitude of the Holocene AHP; there is no coordinated PMIP experiment for the previous AHPs.

We have changed the phrasing in the text.

3. Line 21: I think “confirm” is a strong word since it is well established (and not contested) that precession paces the AHPs. The reported results are, instead, in alignment with what is already known.

We have changed the wording to ‘show’.

4. Line 23: “glacial periods”.

We have changed this.

5. Line 26: The last sentence is out of place. This is not because it is untrue, but because it is completely outside the purview of this study. In fact, the potential implications for the out of Africa dispersal are mentioned only once in the main text, in the last line alone.

We have added additional text to Section 1 discussing the implications of the NAHPs on species’ distributions and evolution.

Suggested improvements

The manuscript needs careful proofreading and some improvements for greater readability. Some suggestions are listed below, but the list is likely not exhaustive:

- Line 38: Inconsistent tense (should use “amplified” and “enhanced”)

We have corrected this in the text.

• Line 47: I believe the authors want to convey that the mechanisms of the NAHPs are poorly constrained, not our ability to understand them.

We have changed the wording in this sentence.

- Line 76: Please indicate a section in the Supplementary Information as “Model validation”.

We have added section numbers to the Supplementary doc and the text.

• Line 143: Please elaborate on the connection between the subtropical westerly jet and the meridional Hadley Circulation.

We have added a description of how the jet is formed via the Coriolis force.

• Lines 159 and 160: The authors should avoid using pronouns for subjects. In Line 159, biogeophysical feedbacks are not enhanced but activated / initiated.

We have changed the wording here

- Line 183: Perhaps the authors want to convey that the magnitude of precession-minima precipitation is not directly correlated with eccentricity forcing.

We want to convey it is not directly correlated to precession forcing, and therefore correspondingly to eccentricity. We have made this clearer in the text.

- Line 194: There is no Figure 3f in this manuscript.

We have changed this in the text.

- Figure 2: Pmax is indicated in the left panel.

This has been changed.

- Figure 4, Extended Figures 3, 4 and 5: Please rephrase the last sentence of the caption.

We have rephrased these sentences.

- All through the manuscript, please use 15 °W instead of -15 °E (for example).

We have changed this in the text.

We have also made a range of other improvements to the text as seen in the tracked changes.

Thank you for your review comments.

Reviewer #3 (Remarks to the Author):

Armstrong et al. present a well-written and potentially critical study on North African humid periods (NAHP) during the past 800,000 years. The overall approach applied by the authors is exciting, but I remain unsure if the results can be considered to be robust. For the sake of transparency, I need to say that I am not a modeler, and thus, reviewing this paper from the perspective of a "proxy-person," not a modeler, which also means that some modeling details are not necessarily apparent to me.

Thank you for your review comments – hopefully we have clarified our approach and the robustness of our model results.

The reason I mainly struggled with this paper is threefold:

Firstly, the model reproduces modern precipitation patterns well enough (Supplementary Figure 1), but it tends to produce more widespread precipitation patterns that are more intense than the compared to the average observation time series (1950-2020). So, it seems the model is always on the "too wet" side for regions it produces precipitation, thus potentially overrepresenting wet phases in some regions.

As with all climate models, precipitation will be subject to regional biases. HadCM3 has always had a wet bias over Africa for the present-day period as pointed out by the reviewer. However, these biases are within the range of more advanced CMIP5 models (see Section 5.1.2 of Valdes et al. 2017). It is because the model is prone to drizzle like almost all GCMs, which impacts hyper-arid regions such as the Sahara. Despite the model being too wet when it its 'standard' configuration, this did not translate into any green periods for the Holocene or earlier. This shows the "standard" version of the model misses all green periods whilst the "paleo-conditioned" version, the focus of our study, is much more successful.

However, within the Sahara this bias is on the order of 0.19 mm/day for the summer (JJA), with an average annual positive bias of 0.06mm/day equivalent to 21 mm/yr (360 model days in a year). This may impact the amplitude of the NAHPs, however this bias is very small compared to the overall amplitude of the NAHPs. Furthermore, it is unlikely to impact the large-scale mechanisms that we propose, both for how precession drives the NAHPs or how the ice sheets act to suppress NAHPs during glacial periods.

We have added to the text and acknowledged that HadCM3 is slightly too wet over Africa for the present day in both Section 1 of the Supplementary information and in Section 3 of the manuscript, and we note that this may amplify the amplitude of the simulated NAHPs. As stated, all models display biases but we do not believe that this impinges on the main findings that we present.

At the same time, the model does not represent specific precipitation patterns shown to have existed during past NAHP based on proxy records. That is particularly true for the entire North-Eastern Sahara region. In all fairness, this model and proxy data mismatch for this region is recognized by the authors (Line 116-127) but not further discussed in detail within the discussion. In my mind, however, this represents a huge cavity of the model. From the way it looks to me, the model does not only underrepresent the precipitation in North-Eastern Africa, as stated by the authors, but it simply does not produce precipitation for half of North Africa even though from proxy data, substantial lakes and active river system have been reconstructed for this region (e.g., Blanchet et al., 2021; <https://doi.org/10.1038/s41561-020-00671-3>).

We agree that this is a major caveat with the model. The model results are not perfect for the North African humid periods, however models will never provide a perfect representation because they will always be subject to some biases due to the inherent approximation of the real system. As we state in the text, this mismatch in N/NE Africa is currently an area of ongoing research for us and may be due to a multitude of factors. As with the initial paleoconditioning of the model, this is a significant challenge to rectify.

We agree that the simulated NAHP precipitation in North Africa, such as the Tibesti mountains, is less than that provided in Blanchet et al., 2021 (360mm/yr compared to their ~500mm/yr). However it is worth noting that the present-day model bias with LOVECLIM in Blanchet et al. is significantly greater than ours for that region (see their Extended Figure 5, which implies a bias of

up to ~1000mm/yr). They do not explicitly state that they bias correct their data in the Methods, so this bias may cause a much larger overestimate of the past precipitation in these regions.

However, we agree that the response in the Northeast and Arabian peninsula is too weak and have added that this is a caveat of the model to the text. We have expanded upon the reasons for this in the last paragraph of section 3. In the final summary paragraph, we have again reiterated that the model does not produce the NAHP response indicated by observations in the NE. We also state that the WAM driver mechanisms that are apparent in the Western Sahara may not be present in the Northeast, and that further work is required to resolve this discrepancy.

Lastly, the authors claim that the model overall produces the NAHP well across the past 800 kyr, including the "skipped beats." However, I need help seeing this proposed match in the Fig. 1j record. Most of the skipped beats are expressed as "glacial sapropels" in proxy records of the Mediterranean and thus are reflected to some extent in the wet-dry index of the Mediterranean Sea (Fig. 1d). Yet, the model, in my view, persistently fails to produce enough precipitation for these time periods. Based on Fig. 1, I am not convinced the model works well for these time periods.

Again, models are not perfect representations of the system and will be subject to uncertainties. However, these uncertainties are also present in the observational proxy data which also show a number of disparities. Therefore there is a degree of subjectivity when interpreting and comparing where the NAHPs occur and their possible amplitudes. We have matched the vast majority of the NAHP events in our model data to distinct peaks in the observation timeseries and the sapropels. This is except for 4 events, which as we state in the text, may be due to uncertainties regarding the ice sheet model. It is difficult to match proxy precipitation reconstructions with climate model simulations, particularly over such a long time period. Therefore, we think that what we have presented is a good comparison between the model and observations, and as such is an advance on earlier work. We have also showed that the modelled NAHPs are of comparable amplitude in the West Sahara to the sparse number of estimates of the Holocene AHP intensity.

The North African proxies that we use in Figure 1 represent moisture availability in the Nile River Basin and across North Africa. Although we might not have enough precipitation in the NE and Arabian Peninsula during the NAHPs, there is still an increase in the Nile River source area and in Western North Africa. Therefore, there is justification to compare our Saharan timeseries with these observations as they cover a very large source region. Also these proxies do not directly track precipitation and the scales are unlikely to be linear with regards to precipitation rate. Therefore it is not possible to judge the actual amount of precipitation in these regions using either the proxy timeseries or sapropels.

Put all the above together, I need help understanding how the model output can be used to assess North African humid periods if the model does not reproduce half of North Africa particularly well while the other half might be overrepresented due to the sensitivity of the model. Additionally, some threshold conditions are seemingly not presented. Hence, I would appreciate it if the authors could revise the manuscript to address these issues, which seems crucial to get a feeling of the robustness of the model output. Currently, and I am sorry to say this, but the output does not convince me.

We agree that the model underrepresents precipitation in the NE Africa and have identified that this is a caveat of the model and potential reasons for it. The direct mechanisms that drive the NAHPs in this region therefore cannot be clarified in this study, we have stated this in the text. The overrepresentation of precipitation in the West due to model bias is on the order of 21mm/yr which is very small compared to the amplitude of the simulated NAHPs, we have also clarified this in the text.

Models will always have inherent biases and uncertainties; however, we believe that this manuscript is an advancement on previous studies due to methodology that we have employed, the ability to simulate any greening of the Sahara in a coupled climate model (which is a very big improvement on previous efforts), the improved model resolution compared to other studies, the length of the time period analysed, and the detail at which we investigate the mechanisms for the NAHPs are formed and how they are suppressed by the ice sheets during glacials.

Generally, the authors could incorporate available proxy records much more. While the authors use proxy records to assess model robustness for surface air and sea surface temperatures

(Supplementary Figures 2 and 3), the cross-validation with proxy records in terms of the precipitation output is not existing. The authors use the wet-dry index from the Mediterranean, the dust record from the West Sahara region, and the δD record from the Gulf of Aden to identify the NAHP. But why these records? Why not use Lake Bosumtwi for West Africa for some of the studied period (Miller et al., 2016; <https://doi.org/10.1002/jqs.2893>)? Why not use Chew Bahir, Lake Magadi, or Lake Tana proxy data to assess NAHP from the eastern African viewpoint (e.g., Foerster et al., 2022; <https://doi.org/10.1038/s41561-022-01032-y>)? The authors used the data for temperature. Why not precipitation? Why use the Gulf of Aden proxy record when it is so short? The selection of records seems random, and I would like the authors to explain why they selected those records and how they can be meaningful to the reader.

Our selection of the proxies in Figure 1 is based on their proximity to the Sahara region, their long timescale and high temporal resolution (i.e. clearly resolving precession). The Gulf of Aden record is δD , which is the most direct proxy of precipitation available for North Africa. Furthermore, it has been very highly cited in the context of NAHPs. We have added an additional paragraph in section 3 which introduces each of the proxy datasets presented in Figure 1 and our justification for using them.

We have not added additional proxies to Figure 1 from Western and Eastern Africa because they are outside the study area. Both West and East Africa are within different climatic regimes, i.e. the tropics or the topographically complex East African rift valley. These regions are likely to be respond differently to orbital forcing compared to the arid Sahara/Sahel. As such we do not think it is appropriate to compare these proxies to a model timeseries of Saharan precipitation as we do in Figure 1, and they do not clarify the mechanisms and conclusions that we present. For example, the Lake Magadi and Lake Tana dataset are of poor resolution and do not show precession, therefore they are not useful in assessing the impact of precession-driven insolation change which drives the NAHPs.

In order to address your request, we have added a precipitation validation in section 1 of the supplementary information. This gives a comprehensive overview of African precipitation proxies, including all of those that you recommend above, and compares against our model data. In most instances, the observations show good agreement with the model data which demonstrates the robustness of our model results.

Regarding the vegetation output, I would also like to invite the authors to provide cross-evaluation based on proxy data. This, too, could be cross-checked by using the NEOTAMA database and CREST (Chevalier et al., 2022; <https://doi.org/10.5194/cp-18-821-2022>). While pollen data for the past 120 kyr is more readily available than for the past 800 kyr, the composite vegetation pattern produced from the 20 NAHP could be cross-referenced with what is available in terms of proxy data.

The paleo-conditioned model has already been evaluated against the mid-Holocene vegetation cover, for which the pollen dataset already exists, in Hopcroft & Valdes (2021) as shown in their Figure 2.

However, even for the Holocene, North African pollen data is sparse in the NEOTOMA database and is basically non-existent for Saharan-wide reconstructions for pre-Holocene periods. Therefore this request would require us to generate a new pollen data product in order to compare with our model study. Generating such a composite of pre-Holocene spatial pollen records would, in our view, constitute a major study in itself that should be done by the pollen- and chronology community. Additionally, CREST is a highly useful statistical palaeoclimate (not vegetation cover) reconstruction technique, especially in the regions, where e.g. modern pollen calibration data are sparse. However, CREST is not useful if no fossil pollen records are available. As a result of this, we think that this request is beyond the scope of this study.

Lastly, the authors state that winter storm tracks, a vital moisture source during winter to North Africa (L116-127), need to be better resolved in the model. The authors still go on to discuss high-low latitude atmospheric teleconnections to explain the precipitation patterns the model provides. How can the mechanism be discussed if an important atmospheric aspect linking high and low latitudes and which is also sensitive to European Ice sheet conditions (e.g., southward extension; Luetscher et al., 2015; <https://doi.org/10.1038/ncomms7344>) is not represented in the model? I would appreciate it if the authors could discuss this aspect. How would that change the results,

assuming that storm tracks have an effect? Why does it not matter that the storm tracks are not represented?

Although storm tracks are represented in the model, modelling them is very complex. Even with the most advanced CMIP5/6 generation of climate models, simulating Mediterranean (and wider extratropic) storm tracks is subject to large intermodal spread and uncertainties (e.g. Priestley et al., 2020). Mediterranean cyclones are commonly driven through interactions with orography, which is influenced by the resolution of our model, e.g. Italy and the Alps are not fully resolved. As such, biases and uncertainties in Mediterranean storm tracks is to be expected, as it is even with the most advanced climate models.

However, storm tracks and the large-scale reorganisation of the WAM and jet streams that we identify occur on significantly different spatial scales. The impact of the storm tracks is primarily isolated to the coastlines during winter, we have changed the text to clarify this. Therefore they are not expected to have a large influence on the amplitude of the NAHPs inland, as is shown by Kutzbach et al. 2020 and Blanchet et. al. 2021. In contrast, the large synoptic scale WAM and jet mechanisms that we propose impact the whole summer monsoon system and are on a much larger scale that effects the Sahara across a wide latitudinal band. Therefore, these large-scale atmospheric changes can be considered to be more robust that the finer scale processes that constitute the storm tracks.

In response to your questions, it is likely that better representation of the storm tracks would increase winter precipitation along the coastlines, we have clarified this in the text. Therefore they do matter, however other modelling studies have shown their impact on the NAHPs to be small relative to the summer monsoon changes (Kutzbach et al. 2020, Blanchet et. al. 2021).

In summary, I am intrigued by the author's findings that CO2 and AMOC changes did not contribute much to the precipitation patterns across Africa. This builds on recent proxy results suggesting something similar (Gosling et al., 2021; <https://doi.org/10.1126/science.abg4618>). The separation of the short eccentricity cyclicity from the precession cyclicity and the subsequent argument that the former stems from Ice sheet variability rather than the low latitude insolation forcing itself is a significant finding. However, I am unsure about the model output as a whole; hence the wavelet results are also difficult to judge for me at this point.

We have added Extended Figure 7 in order to clarify that there is no link between AMOC strength and the NAHPs as has been previously proposed.

We hope that we have emphasised the advancement of this study compared to previous work and the robustness of our model results (and subsequently the wavelet plots).

Thank you again for your review comments.

Reviewer #4 (Remarks to the Author):

Co-reviewed with reviewer 2

Reviewer #2 (Remarks to the Author):

The authors have made several changes to the manuscript, which have enhanced the readability of the manuscript and allow greater appreciation of the key results. I think the point about the influence of eccentricity coming into play through ice-sheet extent and not through modulation of precession has been explained well in the Discussion. This is an important finding and it has been well argued for. I also appreciate the detailed discussions added to the Supplementary Information. I am satisfied with their response to my comments, including those related to the originality and significance of the work.

I have only some minor points to add:

- Line 51: "Most climate models fail to reproduce the precipitation change seen in the Holocene NAHP". This is a good point to mention but seems misplaced in a discussion of NAHPs during glacial periods (the sentences before and after are about glacial periods). The authors may consider placing this elsewhere.
- Line 110: "across North Africa"
- Line 301: "climate dynamics of the past and future"
- Figure 1: From the caption, following the reference for the standardized West African Monsoon index (7), the reader might be led to Larrasoana et al. (2021) in the list of references, instead of Akinsanola et al. (2020) in the extended list of references (which I presume is what the authors actually intended).
- Figure 2: I think the label in the panel 2 should read "Contour from -24 to 24 by 2" but the minus sign is missing. For the readers' ease, the authors should also mention which elements of the plot (colours/contours) represent which variable (vertical velocity/zonal wind) since this may not be immediately clear to everyone.

Reviewer #3 (Remarks to the Author):

Dear Editor,

I very much appreciate the diligent work of Armstrong et al. during the review process and that they endeavored to address and satisfy all the review comments. I also remain steadfast on my view that this paper can potentially be an extremely valuable contribution to our understanding on how past climate change influenced the African monsoonal system and thus drove the availability as well as viability of habitats.

However, my main point of critic was that the model fails to adequately project African Humid periods which are defined as humid periods across the entirety of northern Africa. In fact, the model simply does not produce precipitation across half of northern Africa (eastern Africa + Arabia) and instead is centered almost exclusively across western Africa. In my previous review, I had hoped by pointing out this major cavity, the authors might find a way to change model parameters to address this. The response of the authors, however, clearly states that the model is simply not good enough yet to achieve this.

With this in mind, the authors still argue in the revised version of the manuscript that their findings of timing, spatial distribution changes and forcing mechanism of northwestern Africa precipitation can be applied to the entire North African realm despite the fact it that the model output does not support this. And this is exactly where I see the problem if this publication gets accepted at it currently stands. This generalization based only on northwestern Africa but transferred to the entirety of northern Africa in order to be about African Humid Periods as a whole carries a huge uncertainty. How can the authors be sure that the timing and particularly forcing sensitivity they found is not just limited to the western African monsoonal system? Previous

studies have made exactly the argument that the western and eastern monsoonal system in Africa did not necessarily hold the same timing or sensitivity to orbital forcing (e.g., Trauth et al., 2009; QSR). Additionally, several papers in the past couple of years have argued that ice volume changes on orbital time scales had no impact on African monsoonal changes (e.g., Trauth et al., 2021; QSR), and that particular (north)eastern Africa is more susceptible to orbital paced changes in the Walker Circulation than global ice volume (e.g., van der Lubbe et al., 2021; Nature; Foerster et al., 2022; Nature Geoscience; Lepre and Quinn, 2022; Global and Planetary Change; etc.). As the authors cannot analyze (north)east vs. (north)west based on their model output this issue will remain unanswered for now.

While I agree with the authors that this model is better than its predecessors and appreciate the fact that a model can never completely capture Nature's complexity, the question that needs to be asked here is: why do the authors insist on selling this paper on something, this being African Humid Periods as a whole, when their model data cannot deliver on it?

My recommendation to move forward is, that the authors not try and sell this paper for something it is not. This paper provides crucial and valuable insights into (north)western African monsoonal system. A region that is chronically underrepresented by proxy data. These insights could be invaluable for future paleoclimatic reconstructions, and thus should be made public after another round of reviews. I would strongly recommend that the authors remove the phrase "African Humid Period" from the entire manuscript incl. the title, and replace it with something the model actually produces which is "western African wet phases" or something else along these lines that the authors favor. I would also recommend the authors implement a sentence that their findings might not be represent the northeastern African realm as the model does not capture this region. Hence, their findings, in particular to the influence of global ice volume changes, need to be tested against model or proxy data from the eastern African realm in the future.

REVIEWER COMMENTS

Reviewer #2 (Remarks to the Author):

The authors have made several changes to the manuscript, which have enhanced the readability of the manuscript and allow greater appreciation of the key results. I think the point about the influence of eccentricity coming into play through ice-sheet extent and not through modulation of precession has been explained well in the Discussion. This is an important finding and it has been well argued for. I also appreciate the detailed discussions added to the Supplementary Information. I am satisfied with their response to my comments, including those related to the originality and significance of the work.

Thank you again for your review comments.

I have only some minor points to add:

- Line 51: “Most climate models fail to reproduce the precipitation change seen in the Holocene NAHP”. This is a good point to mention but seems misplaced in a discussion of NAHPs during glacial periods (the sentences before and after are about glacial periods). The authors may consider placing this elsewhere.

We have reworded this sentence.

- Line 110: “across North Africa”

This has been changed

- Line 301: “climate dynamics of the past and future”

This has been changed

- Figure 1: From the caption, following the reference for the standardized West African Monsoon index (7), the reader might be led to Larrasoña et al. (2021) in the list of references, instead of Akinsanola et al. (2020) in the extended list of references (which I presume is what the authors actually intended).

We have added the Figure captions to the end of the manuscript document in order to integrate the Figure caption references into the main reference list. This is now complete and in the correct order.

- Figure 2: I think the label in the panel 2 should read “Contour from -24 to 24 by 2” but the minus sign is missing. For the readers’ ease, the authors should also mention which elements of the plot (colours/contours) represent which variable (vertical velocity/zonal wind) since this may not be immediately clear to everyone.

That has been corrected. We have further described each element of the plot (i.e. the

colour contour and line contours) in the Figure caption so this should be very clear to the reader. We have not added this to the main Figure due to space requirements – adding further text makes an already busy plot look too cluttered.

Reviewer #3 (Remarks to the Author):

Dear Editor,

I very much appreciate the diligent work of Armstrong et al. during the review process and that they endeavored to address and satisfy all the review comments. I also remain steadfast on my view that this paper can potentially be an extremely valuable contribution to our understanding on how past climate change influenced the African monsoonal system and thus drove the availability as well as viability of habitats.

However, my main point of critic was that the model fails to adequately project African Humid periods which are defined as humid periods across the entirety of northern Africa. In fact, the model simply does not produce precipitation across half of northern Africa (eastern Africa + Arabia) and instead is centered almost exclusively across western Africa. In my pervious review, I had hoped by pointing out this major cavity, the authors might find a way to change model parameters to address this. The response of the authors, however, clearly states that the model is simply not good enough yet to achieve this.

With this in mind, the authors still argue in the revised version of the manuscript that their findings of timing, spatial distribution changes and forcing mechanism of northwestern Africa precipitation can be applied to the entire North African realm despite the fact it that the model output does not support this. And this is exactly where I see the problem if this publication gets accepted at it currently stands. This generalization based only on northwestern Africa but transferred to the entirety of northern Africa in order to be about African Humid Periods as a whole carries a huge uncertainty.

How can the authors be sure that the timing and particularly forcing sensitivity they found is not just limited to the western African monsoonal system? Previous studies have made exactly the argument that the western and eastern monsoonal system in Africa did not necessarily hold the same timing or sensitivity to orbital forcing (e.g., Trauth et al., 2009; QSR). Additionally, several papers in the past couple of years have argued that ice volume changes on orbital time scales had no impact on African monsoonal changes (e.g., Trauth et al., 2021; QSR), and that particular (north)eastern Africa is more susceptible to orbital paced changes in the Walker Circulation than global ice volume (e.g., van der Lubbe et al., 2021; Nature; Foerster et al., 2022; Nature Geoscience; Lepre and Quinn, 2022; Global and Planetary Change; etc.). As the authors cannot analyze (north)east vs. (north)west based on their model output this issue will remain unanswered for now.

While I agree with the authors that this model is better than its predecessors and appreciate the fact that a model can never completely capture Nature's complexity, the question that needs to be asked here is: why do the authors insist on selling this paper on something, this being African Humid Periods as a whole, when their model data cannot deliver on it?

My recommendation to move forward is, that the authors not try and sell this paper for something it is not. This paper provides crucial and valuable insights into (north)western African monsoonal system. A region that is chronically underrepresented by proxy data. These insights could be invaluable for future paleoclimatic reconstructions, and thus should

be made public after another round of reviews. I would strongly recommend that the authors remove the phrase “African Humid Period” from the entire manuscript incl. the title, and replace it with something the model actually produces which is “western African wet phases” or something else along these lines that the authors favor. I would also recommend the authors implement a sentence that their findings might not represent the northeastern African realm as the model does not capture this region. Hence, their findings, in particular to the influence of global ice volume changes, need to be tested against model or proxy data from the eastern African realm in the future.

We thank Reviewer 3 for their comments.

The first component of this review concerns the bias in modelled precipitation in NE Africa/Arabia during the NAHPs, and the implications this has on our proposed mechanisms. We agree that the model most likely underperforms in NE Africa and Arabia. We have realised this since the beginning of our work and made this clear in the first version of the manuscript. We have added further text regarding this caveat in a number of locations following this review. Saying this however, there remains uncertainty regarding the extent of this bias, as proxy observations are sparse and unambiguous, and wetter conditions are not always local in origin. We discuss this in greater detail below.

We also agree that the mechanisms that drive the NAHPs may not be entirely the same in Northeast and Northwest Africa. Again, we have made this explicit in the manuscript and have added further clarification following this review. However, the model and proxies produce the same pattern (timing and relative amplitude difference between arid and humid periods) in both western and eastern halves of North Africa, which suggests that similar forcing mechanisms are in effect across N Africa. The difference being that over NE Africa/Arabia, the model does not produce the expected amount of precipitation. Again, we discuss this in greater detail below.

In light of the biases in the model, the second point in the review recommends relabeling our modelled precipitation events, from the established ‘(North) African Humid Period’ nomenclature. We believe that this is a step too far, and that it is counter-intuitive to rename these paleoclimate events based on a potential model bias. As we discuss below, there remains significant uncertainty regarding the amplitude of this potential bias in NE Africa, due to the paucity and uncertainty with proxy observations. Even though the model may be underperforming in this region, the orbitally driven events that the study is focused on still constitute the African Humid Periods, albeit of potentially lower amplitude in the NE. While sometimes debated, the ‘(North) African Humid Period’ nomenclature is a well-established term and has been used in numerous past modelling studies that have produced a much weaker, and spatially heterogeneous precipitation response than our model has (e.g. Meniel et al. 2021; Duque-Villegas et al. 2022; Timm et al. 2010). The term is also used in proxy-based studies relying often only one proxy record. Therefore, we believe there is a strong rationale to continue to use this term in this study. Introducing new terminology that will not be recognised by readers does not improve this study or facilitate better understanding of precipitation variability in North Africa.

In order to justify this, we address the review points in more detail here. Firstly, the suggestion that the model is only able to simulate precipitation in western Africa is an exaggeration of the bias. When N Africa is divided in the Western and Eastern halves (along 15°E), the average precipitation in Eastern Africa during the NAHPs (c. 200mm/year) is still enough to produce greening in the area (Figures 2–3, extended data figure 2). It is only the NE corner of the Africa (Egypt, E. Libya and N. Sudan) and Arabia where the model may fail to produce the expected amount of precipitation.

Saying this however, the model-data discrepancy in NE Africa remains highly uncertain. Proxy data from N. Sudan, Egypt, and E. Libya (the region where modelled precipitation remains very low) are far from being unambiguous regarding the hydroclimatic state of the region during the NAHPs. Eastern Mediterranean proxies tracking river discharge are influenced by precipitation in the Central Saharan watershed (e.g. Tibesti and Tassili n'Ajjer) and Nile source areas (Osborne et al. 2008; Grant et al. 2017, Revel et al. 2010) that are within the humid regions in the model. Large paleo lakes like Lake Megafazzan had a catchment in the areas where the model shows increased precipitation (Tassili n'Ajjer; Drake et al 2008). Similarly, in Libya an important fraction of available water is thought to have come through paleo rivers, and not as a local precipitation (Drake et al. 2008, Osborne et al. 2008). Groundwater also had an important role in the maintenance of wetlands and could have been recharged distances away from the location of a water body and precipitation events (Lézine et al. 2011; Nicoll 2004; Kieniewicz and Smith 2009; Jahns 1995).

Furthermore, the precipitation signal is not consistent across different studies. Some proxies indicate that during the Holocene NAHP, NE African lakes, swamps, and playas existed at 17°N (400mm in the model), savanna grasslands between 20°N and 22°N, while above 22°N, rainfall sustained only patches of grasses (Nicoll 2004). This indicates generally arid conditions in the eastern Sahara above 22°N, i.e. in the area where the model fails to produce precipitation. Proxy data compiled by Lézine et al. (2011) shows the persistence of locations with arid conditions in the Egyptian sector of the Sahara throughout the Holocene NAHP. Pollen evidence is relatively sparse (Prentice et al 2000; Harrison, 2017), only one site shows that grasslands encroached into Arabia during the Holocene AHP, and the northernmost site shows desert persisted at around 25°N (Harrison et al 2017). Furthermore, pollen data also suggests that tropical forest and wooded grassland components entered the desert along rivers and lakes where they benefited from permanent fresh water, while Saharan trees and shrubs persisted outside the water bodies (Watrín et al. 2009). This indicates that water availability away from rivers and lakes may have been much reduced.

Therefore, the increased availability of water during humid periods seen in a number of proxy records in the NE corner of Africa does not necessarily mean high local precipitation. There is no reason to assume equal distribution of precipitation across N Africa and no clear evidence exists showing that the NE corner of Africa should have local precipitation levels as high as in the central and western Sahara. Therefore, while we are ready to acknowledge that the model may not simulate sufficient precipitation in NE Africa, the discrepancy isn't necessarily as big as is being argued here.

Secondly, R3 suggests that the western and eastern monsoonal system in Africa did not necessarily hold the same timing or sensitivity to orbital forcing. We have clarified in the paper

that the mechanisms we identify as driving the NAHPs and skipped beats in the Western Sahara may not be prevalent in the NE. Saying this however, the proxy record relevant for the Eastern Sahara (Grant et al. 2017) clearly shows similar timings and (relative) amplitude changes during humid periods compared to the western part of N Africa (Crocker et al. 2022). In addition, paleo-lake and limestone formation data from Libya and Egypt show a clear 100kyr cyclicity, where humid phases coincide with NH interglacials (Drake et al. 2008; Geyh & Thiedigh 2008; Crombie et al. 1997, Szabo et al. 1995). This shows that the timing of humid phases is similar across N Africa and follows the eccentricity-controlled ice volume cyclicity. This supports our results and the idea that the same mechanisms were influencing eastern and western Saharan humidity, even if there may have been some additional, currently unknown, factors influencing the overall humidity in the NE Sahara.

In order to question the forcing sensitivity between West and East Africa, four studies are presented - Trauth et al. 2009, van der Lubbe et al., 2021; Foerster et al., 2022 and Lepre and Quinn, 2022. However, other than Trauth et al. (2009) none of these studies reflect climate in the Eastern Sahara (20-40°E, 15-35°N), instead they record climate of equatorial (sites at around 4°N) or southern equatorial Africa (one records from the Mozambique channel at 15°S). Therefore, we should not expect these records to have the same timing and forcing sensitivity of humid periods as the Western Sahara. For example, Shanahan et al., (2015) showed that the end of the latest AHP was relatively synchronous across the Sahara, but these sites all persisted in a wet state for another 2-4 kyr afterwards. This is all supported to some extent by our model results for these regions which are compared with Chew Bahir (Trauth et al., 2021) and other records from equatorial and Southern Africa in the Supporting Information (Figure S4). These comparisons already show that the model behaves differently in equatorial Eastern and Southern Africa, probably because of other influences like the Walker Circulation, as is pointed out.

It is clear therefore that in the proxy record there remain significant uncertainties regarding the amplitude of the NAHPs in NE Africa, and that proxies indicate a similar W-E temporal pattern of the NAHPs, in agreement with our model. Therefore, we do not think there are grounds to re-label these events from the standard '(North) African Humid Period' terminology. These are orbitally driven events that in numerous other studies are referred to using this term. Introducing new terminology is unnecessary, will confuse readers, and does not improve our study or enhance our understanding of African precipitation variability.